# Mutation in the rat interleukin 34 gene impacts macrophage development, homeostasis, and inflammation

Stephen Huang[1], Omkar L Patkar[1], Sarah Schulze[1], Dylan Carter-Cusack[1], Susan Millard[1], Ginell Ranpura[1], Emma K Green[1], Emma Maxwell[1], Jeeva Kanesarajah[5], Gary Cowin[2], Damion Stimson[3], Nyoman D Kurniawan[3], Sahar Keshvari[1], Rachel Allavena[4], Allison R Pettit[1], Katharine M Irvine[1], David A Hume[1]

Interleukin 34 (IL34) and colony-stimulating factor 1 (CSF1) signal through a shared receptor (CSF1R) to control macrophage survival, differentiation, and function. Here, we describe the impact of loss-of-function mutation in the rat *Il34* gene. In contrast to IL34 mutant mice, macrophages within squamous epithelia (Langerhans cells) were not significantly depleted in *Il34*$^{-/-}$ rats. In the brain, microglia and brain-associated macrophages were selectively depleted in grey matter. A gradient of microglial density in the *Il34*$^{-/-}$ cortex suggests that CSF1 can diffuse outwards from the corpus callosum. Microglial loss was not associated with detectable neuropathology or altered gene expression in the cortex, hippocampus, and thalamus aside from selective loss of microglia-expressed transcripts. In the adenine diet model of renal interstitial fibrosis, both *Il34* and *Csf1* were induced. The absence of IL34 led to a significant reduction in macrophage recruitment compared with controls, but pathology was unaffected. We suggest that IL34 and CSF1 provide overlapping signals to sustain microglia and to direct macrophage recruitment and repair tissue injury in the periphery.

## Introduction

The mononuclear phagocyte system (MPS) is family of cells that includes committed haematopoietic progenitors, blood monocytes, and tissue-resident macrophages that populate every organ in the body ([1], [2], [3], [4], [5]). The proliferation, differentiation, function, and survival of cells of the MPS are regulated by signals from the macrophage–colony-stimulating factor receptor (CSF1R). CSF1R expression is entirely restricted to MPS cells, and a knock-in transgenic reporter in mice highlights their abundance, location, and regular distribution in every organ ([6]). Homozygous loss-of-function mutations in the *CSF1R* locus in mice, rats, and humans are associated with depletion of MPS cells in most organs, including microglia in the brain and bone-resorbing osteoclasts, and consequent severe pleiotropic impacts on postnatal growth and development including osteopetrosis ([7], [8]). Tissue-resident macrophages remain CSF1R-dependent in adults and can be depleted by treatment with a blocking anti-CSF1R antibody ([9]).

CSF1R has two alternative ligands, CSF1 and IL34. Null mutations in the *Csf1* gene in mice and rats are associated with tissue macrophage deficiency and osteopetrosis ([10], [11]). The existence of a second CSF1R ligand was inferred from the age-dependent recovery of osteoclast function in CSF1-deficient mice and the more severe phenotype observed in *Csf1r*$^{-/-}$ mice ([12]). Human IL34 was identified in 2008 ([13]). Subsequent studies revealed the functional conservation and co-evolution of the two CSF1R ligands in birds ([14], [15]) and fish ([16], [17], [18], [19]) and defined the molecular basis of binding of the two ligands to the receptor ([20]). Although there have been several studies claiming differences in signalling outcome in response to the two factors in vitro ([21]), they appear functionally equivalent in vivo. The transgenic expression of IL34 from the *Csf1* promoter was sufficient to rescue all macrophage-associated phenotypes in CSF1-deficient mice ([22]). In adult mice, *Il34* mRNA is expressed primarily in the brain and epidermis and *Il34*$^{-/-}$ mice have a selective loss of microglia and Langerhans cells ([23], [24]). IL34 has also been shown to regulate microglial colonization of the brain in zebrafish ([16], [18], [19]). The two ligands exhibit cell-specific and spatially restricted expression in mouse brain ([25]); accordingly, administration of anti-IL34 antibody depleted microglia in grey matter, whereas anti-CSF1 antibody affected white matter microglia ([26]). Based upon conditional deletion of *Csf1* using nestin-cre, Kana et al ([27]) suggested that CSF1 was required specifically to maintain

[1]Mater Research Institute-UQ, Translational Research Institute, Brisbane, Australia    [2]National Imaging Facility, Centre for Advanced Imaging, Australian Institute for Bioengineering and Nanotechnology, The University of Queensland, Brisbane, Australia    [3]Centre for Advanced Imaging, Australian Institute for Bioengineering and Nanotechnology, The University of Queensland, Brisbane, Australia    [4]School of Veterinary Science, The University of Queensland, Gatton, Australia    [5]QIMR Berghofer Medical Research Institute, Brisbane, Australia

Correspondence: Katharine.irvine@uq.edu.au; David.hume@uq.edu.au

cerebellar microglia, which were required to regulate cerebellar development. Aside from CSF1R, IL34 has been shown to bind at least three additional putative receptors, PTPRZ1 (28), SYNDECAN1 (29), and TREM2 (30). However, at this time no biological activity of IL34 in vivo has been demonstrated to occur through receptors other than CSF1R.

The increased expression of IL34 has been detected in a wide range of inflammatory diseases and malignancies in patients (21, 31, 32, 33, 34, 35, 36, 37, 38, 39, 40, 41, 42, 43), but it remains unclear whether expression exacerbates or mitigates pathology. One group has proposed that IL34 produced by regulatory T cells has a role in transplant tolerance (44, 45). IL34 in mice has been implicated as an inducible mediator of tissue damage in acute kidney injury (46) and in a lupus nephritis model (47). Conversely, rat models of renal injury have emphasized the inducible expression of CSF1 as a mediator (48, 49). Although these studies have highlighted CSF1R as an anti-inflammatory drug target, CSF1 treatment actually promoted tissue repair in kidney ischaemia/reperfusion injury (50).

Rats have many advantages over mice as a model of human physiology and pathology (51). In terms of CSF1R biology, there are clear differences in the impact of *Csf1* mutations between species. Unlike *Csf1*$^{op/op}$ mice, *Csf1*$^{tl/tl}$ rats have an unremitting osteoclast deficiency and osteopetrosis but somatic growth and male fertility are not affected (reviewed in reference 51). In contrast, homozygous mutation of the rat *Csf1r* gene has a severe impact on postnatal growth and development (52, 53). These observations suggested to us that the precise regulation and function of IL34 in control of MPS development, homeostasis, and immunopathology may differ between mammalian species. To investigate this hypothesis, we created a null mutation in the *Il34* gene in the rat germ line. Analysis of the brain of *Il34*$^{-/-}$ rats confirmed the selective loss of grey matter microglia observed with anti-IL34 treatment in mice and provided evidence of haploinsufficiency. Gene expression profiling of multiple brain regions indicated that there is little distinction between grey and white matter microglia, or selective gene regulation by IL34. Based upon reported roles of IL34 in kidney inflammation in mice, we compared the response of WT and *Il34*$^{-/-}$ rats in a model of chronic renal injury. We conclude that IL34 induction contributes to the recruitment of macrophages in response to epithelial damage, but the absence of IL34 had no detectable effect on either initial injury or repair. We suggest that IL34 is unlikely to be a target for anti-inflammatory therapy.

# Results

## Transcriptional regulation of Il34 in rats and humans

Homozygous mutation of *Il34* in mice leads to selective depletion of microglia in the brain and Langerhans cells in the epidermis (23, 24). Consistent with this pattern, amongst mouse tissues, *Il34* is most highly expressed in the brain and skin (see BioGPS.org). Genome-scale 5'RACE by the FANTOM5 consortium revealed that expression in these two locations is driven by alternative tissue-specific promoters (54). *Il34* mRNA is more widely and uniformly expressed in human tissues (see consensus data on www.proteinatlas.org) again involving

separate transcription start sites (TSS) (54). In humans, a third TSS is used by a wide range of mesenchymal cell types. *Il34* mRNA is detected in spleen in both humans and mice but not in lymphocytes in any state of activation (54). We recently generated a transcriptional atlas for the rat by integrating 6700 RNA-seq datasets in the public domain (55). We also generated a large RNA-seq dataset comparing gene expression in juvenile WT and *Csf1rko* rats (56). As in humans, *Il34* mRNA is detected at comparable levels in most tissues in both rat datasets (Fig S1A). To determine whether alternative promoter use was conserved, we mapped RNA-seq reads from the entire rat atlas dataset to the rat genome as shown in Fig 1A. This analysis indicated that as in mouse and human, neuronal tissues initiate transcription immediately 5' of the first coding exon (CDS exon 1). In common with human, there is an alternative TSS associated with mesenchymal tissues. Skin tissue was poorly represented in the rat atlas, but transcripts were also detected coincident with the skin-specific 5' non-coding exon. This exon is annotated on Ensembl, and the sequence of the mouse skin–specific *Il34* promoter is conserved in rat and human genomes.

## Phenotypic analysis of Il34$^{-/-}$ rats

Since we commenced this project, Freuchet et al (57) reported analysis of homozygous *Il34* frameshift mutation in exon 3 on an outbred Sprague Dawley background. Their focus was on the proposed function of IL34 in control of autoimmunity. They reported normal postnatal growth, reduced microglia in the hippocampus, evidence of minor liver injury (marginal increase in circulating ALT, but not AST or alkaline phosphatase), the presence of detectable autoantibodies, and increased circulating CSF1. They also reported a selective reduction in CD8$^+$ T cells in the spleen.

The *Il34*$^{-/-}$ rats on the inbred Fischer 344 background were generated by deletion of the 134-bp second coding exon using CRISPR-Cas9 (Fig S1F). This deletion removes 35 amino acids of the active IL34 protein and introduces an in-frame stop codon at the start of the next exon. *Il34*$^{-/-}$ rats were born at the expected Mendelian ratios. Although *Il34* mRNA is highly expressed by osteoblasts (55), as reported previously (57), there was no significant impact of the *Il34* mutation on bone density detectable by micro-computed tomography (Fig 1B), in contrast to the severe unremitting osteopetrosis observed in *Csf1*$^{tl/tl}$ rats (11). Unlike *Csf1r*$^{-/-}$ (52) and *Csf1*$^{-/-}$ (11) rats, postnatal somatic growth was not affected in *Il34*$^{-/-}$ rats (Fig 1C), male-specific growth advantage after sexual maturity was preserved, and both males and females were fertile. In the *Csf1r*$^{-/-}$ liver, Kupffer cells are reduced by around 70% and the pericapsular macrophage population is absent (52). We found no significant impact of IL34 deficiency on either liver population. In contrast to mouse, *Il34* mRNA is highly expressed in rat spleen (55). *Csf1r*$^{-/-}$ rats have a partial loss of red pulp macrophages in the spleen and complete absence of marginal zone (CD169$^+$) and marginal metallophilic (CD209B$^+$) populations, which are CSF1-dependent in mice (53). Unlike the *Csf1r*$^{-/-}$ spleens, splenic white matter topology and the location and abundance of IBA1$^+$ and CD209b$^+$ macrophage populations were unaffected by the lack of IL34 (Fig 1D and E). Given the widespread expression of *Il34* mRNA, we considered whether the absence might influence CSF1

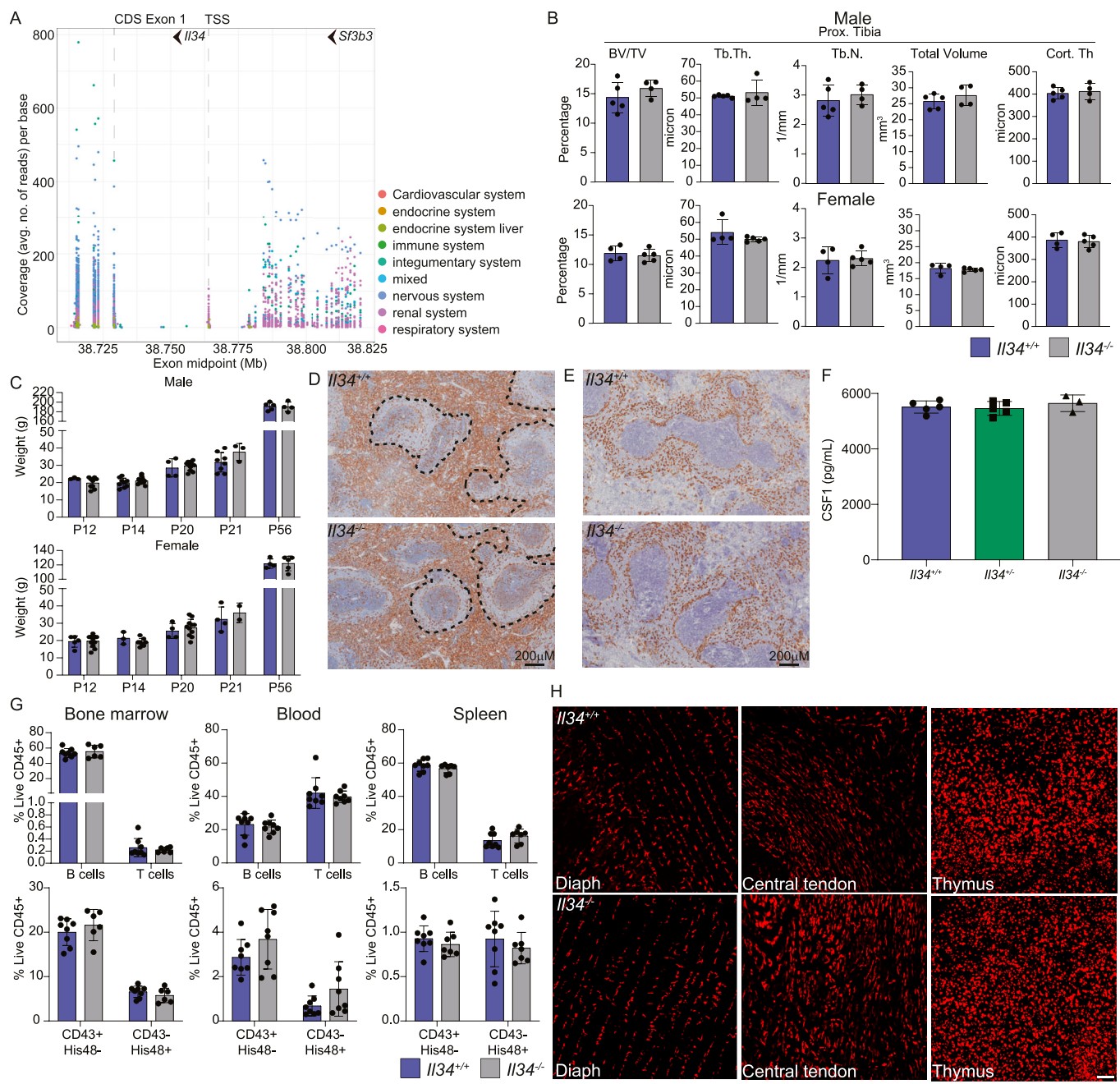

**Figure 1.** *Il34* **is widely expressed in rat tissues but is not required for normal growth and development.**
**(A)** Image shows mapping of RNA-seq reads from the rat atlas to the *Il34* locus. Each dot represents a read count from an individual tissue/library, colour-coded based upon the system as indicated. The location on Chr19 (mRatBN7.2 assembly) is shown on the x-axis; *Il34* is transcribed on the reverse strand; the first coding exon at 38.729 Mb is detected in libraries from all tissues (CDS exon 1). An alternative non-coding exon at ca. 38.753 Mb is detected primarily in libraries from mesenchymal tissues (respiratory, cardiovascular, renal). The most distal 5′ non-coding exon, at 38.764 Mb, is detected at low levels in multiple tissues. To the right of the image, reads are mapped to exons of the upstream *Sf3b3* locus. **(B)** Tibial bone density was measured by micro-CT in male and female rats at 8 wk of age (BV/TV, bone volume fraction; Tb.Th, trabecular thickness; Tb.N, trabecular number; TV, total volume; Cort. Th, cortical thickness). Each dot represents an individual animal (n ≥ 4). **(C)** Body weight gain. WT (blue) and *Il34*$^{-/-}$ (grey) rats were weighed when ear-notched (P12/P14), at weaning (P20/P21) and at P56. Each dot represents an independent animal. **(D, E)** Representative images of spleen morphology and distribution of macrophages in *Il34*$^{+/+}$ (upper panels) and *Il34*$^{-/-}$ (lower panels) rats. IBA1 staining (left) highlights the macrophages of the red pulp, and CD209b (right), the marginal zone macrophages. **(F)** Serum CSF1 was measured by ELISA. Each dot represents an individual animal. **(G)** Flow cytometry analysis of CD45R⁺ B and CD3⁺ T cells, granulocytes, and HIS48⁺(classical) and CD43⁺(non-classical) monocytes from bone marrow, blood, and spleen in 8-wk-old rats (n ≥ 6). **(H)** Ex vivo imaging of diaphragm muscle and central tendon and the thymus isolated from *Il34*$^{+/+}$ and *Il34*$^{-/-}$ animals carrying the *Csf1r*-mApple transgene. Tissues were directly imaged on an Olympus FV3000 confocal microscope with a 10x objective, and scale bar represents 100 μM.

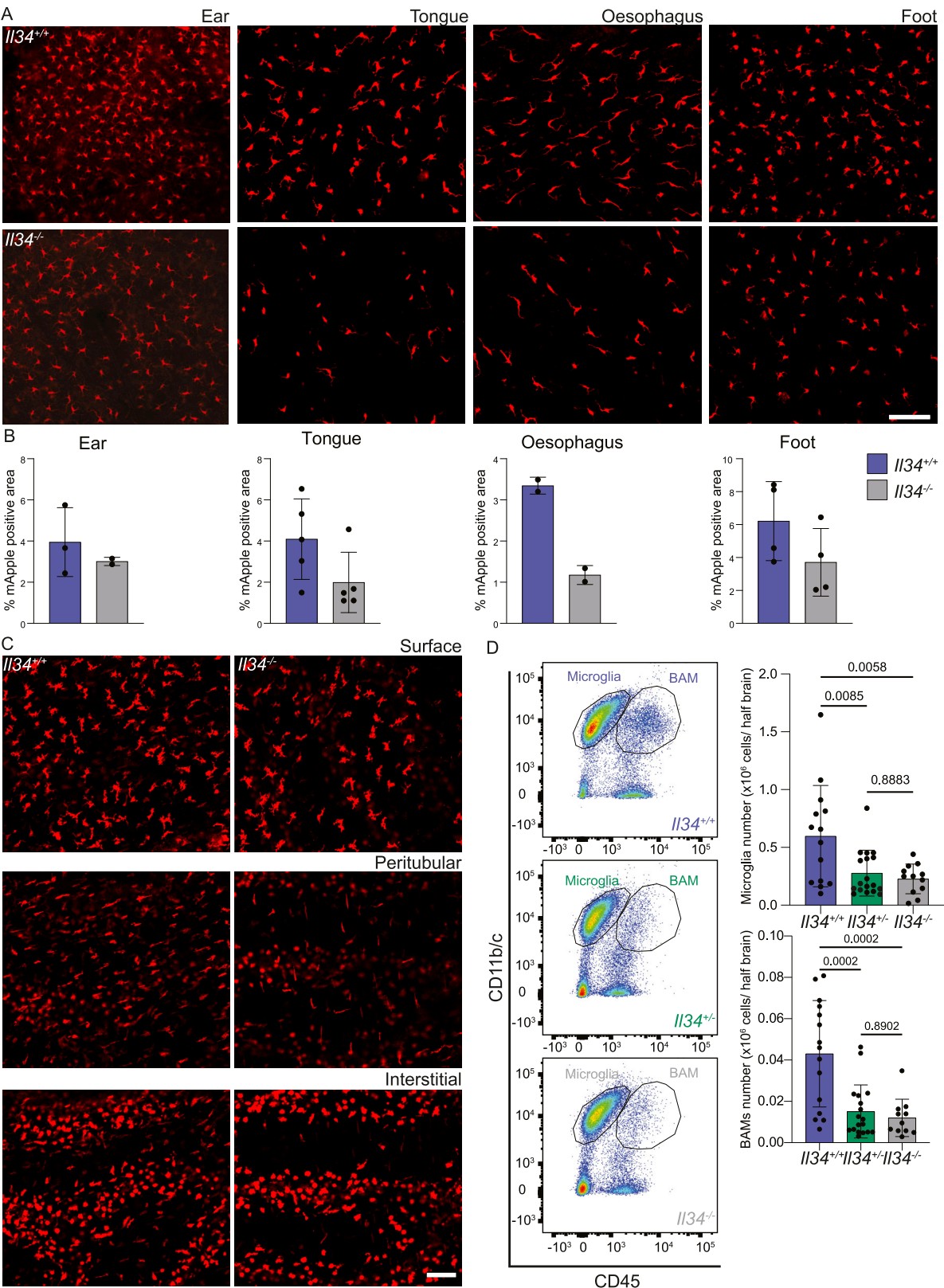

**Figure 2. Loss of IL34 reduces tissue-resident macrophages in squamous epithelia and brain.**
**(A)** Whole-mount imaging of the epithelial layer of the ear, tongue, proximal oesophagus, and foot from *Il34⁺/⁺* and *Il34⁻/⁻* animals carrying the *Csf1r*-mApple transgene.
**(B)** Quantification of percentage area of mApple (RFP)-positive signal in ear, tongue, oesophagus, and foot from IL34⁺/⁺(blue) and IL34⁻/⁻ (grey) animals carrying the Csf1r-

homeostasis. Circulating CSF1 is regulated primarily by receptor-mediated endocytosis and degradation by macrophages and is elevated in $Csf1r^{-/-}$ rats ([52]). However, serum CSF1 levels were not altered in $Il34^{-/-}$ rats (Fig 1F).

Analysis of bone marrow, blood, and spleen by flow cytometry revealed no significant differences in the relative frequency of T cells, B cells, blood monocytes, or granulocytes (Figs 1G) in $Il34^{-/-}$ rats compared with WT. In contrast to the previous analysis of IL34-deficient rats ([57]), $Il34^{-/-}$ rats had a normal distribution of B cells, and CD4 and CD8 T cells in lymphoid organs, including the spleen, thymus, and cervical lymph node (Fig S1B–E). These data indicate that the B-cell deficiency and granulocytosis seen in the rat $Csf1r$ mutation ([52]) are also primarily due to loss of CSF1 rather than IL34 signalling.

To assess the effect of the IL34 knockout on tissue macrophage populations, we crossed the mutation to the $Csf1r$-mApple transgenic reporter line, which is expressed in all tissue macrophages and at lower levels in granulocytes and B cells ([58, 59]). No gross differences in the distribution of tissue macrophages were observed by whole-mount imaging in any organ, including those in which $Il34$ mRNA is highly expressed such as the connective tissue, gut, muscle, and thymus ([55]) (Figs 1H and S1E). In contrast to the reported loss of Langerhans cells in $Il34^{-/-}$ mice ([23, 60]), in $Il34^{-/-}$ rats we could not demonstrate a significant difference in the density of Langerhans cells in squamous epithelia of the ear, tongue, and foot by quantification of *en face* images of the reporter transgene (Fig 2A and B).

Aside from the oesophagus and brain (see below), the only location where we detected a clear loss of a macrophage subpopulation was the testis, which contains three topologically distinct populations, interstitial and peritubular macrophages as described in the mouse ([61, 62, 63]) and a distinct surface/subcapsular population. The surface and peritubular populations were relatively depleted in $Il34^{-/-}$ rats (Fig 2C). Their loss is more readily appreciated through visualization of 3-dimensional z-stacks (Video 1 and Video 2).

### The effect of Il34 mutation on microglial density, location, and gene expression

The brain contains two distinct populations of mononuclear phagocytes, classical microglia and brain-associated macrophages (BAM) ([64]) that line the vasculature and brain surfaces. Flow cytometry analysis of leukocyte populations in disaggregated brain revealed that microglia (CD45$^{low}$/Cd11b/c$^{high}$) and BAM (CD45$^{high}$/CD11b/c$^{low}$) were both significantly reduced in the $Il34^{-/-}$ brain (Fig 2D). The relative abundance of both populations detected by flow cytometry was also reduced significantly in heterozygotes ($Il34^{+/-}$).

We next performed immunofluorescence staining on free-floating sections to assess the morphology and precise location of microglia in specific brain subregions. Figs 3A and S2B show representative images, and Figs 3B and S2C show quantitative analysis of IBA1$^+$ cells in WT, $Il34^{+/-}$, and $Il34^{-/-}$ brains. There is clear

depletion of classical highly ramified microglia in defined grey matter regions in IL34-deficient brain. Interestingly, although the global impact of IL34 deficiency in the cerebellum and brain stem is less obvious, microglia were clearly reduced. Consistent with the analysis in Fig 2, Fig 3 shows that microglial density in multiple brain regions of $Il34^{-/-}$ rats was significantly reduced compared with WT. In this analysis, which has a greatly reduced interindividual variance compared with flow cytometry, the microglial density in every brain region in heterozygotes ($Il34^{+/-}$) was reduced and was intermediate between WT and homozygous mutant. The microglia in heterozygotes remained evenly spaced but at a lower density. Fig 4A shows the density of microglia in the multiple layers of the cortex from the corpus callosum to the meninges in the three genotypes. In the $Il34^{-/-}$ brain, we noted an apparent outward gradient of microglial and perivascular macrophage loss.

To confirm the conclusion from the flow cytometry profile that microglia and BAM were equally affected by the loss of IL34, we located the BAM marker CD163 ([65]) on sections of cortex from wild-type and $Il34^{-/-}$ brain. Fig 4B–F shows the selective loss of CD163$^+$ cells from the grey matter regions where microglia were also reduced but not from the meninges. Taken together, the data demonstrate that IL34 and CSF1 are only partly compartmentalized within the brain, each contributing to the local sustainable CSF1R-dependent mononuclear phagocyte population.

To analyse the molecular consequences of microglial deficiency, we profiled gene expression in the cortex, hippocampus, and thalamus of WT and $Il34^{-/-}$ rats at 10 wk of age. The data are provided in Table S1. Surprisingly, $Il34$ mRNA is still detectable, reduced by around 50% in the mutant rats. Mapping of the primary RNA-seq data to the rat genome confirmed the removal of the second coding exon and generation of a spliced non-coding transcript that presumably does not undergo nonsense-mediated decay.

Consistent with the loss of IBA1 staining in sections, $Aif1$ mRNA was reduced by 40% in the hippocampus and thalamus and 60% in the cortex. As in previous analysis of brain regions in $Csf1r^{-/-}$ rats ([66]), the complete dataset was subjected to network analysis using *Graphia* to identify clusters of co-expressed transcripts (Table S1). The only cluster clearly associated with the IL34 genotype, Cluster 18, contains 80 transcripts including $Aif1$ and $Csf1r$ and overlaps with the microglial signature identified previously in comparison with WT and $Csf1r^{-/-}$ rats ([66]). Fig 5A shows the average expression of genes within this cluster, reduced by 40–60% in each brain region, which parallels the loss of $Aif1$ and IBA1. We did not detect the expression of markers of damage-associated microglia (e.g., $Clec7a$) or the loss of homeostatic markers reported in $Il34^{-/-}$ mice ([27, 67] Preprint). Fig 5B shows the detection of individual transcripts, confirming that homeostatic markers such as $P2ry12$ and $Tmem119$ were reduced to the same extent as $Aif1$, $Csf1r$, $C1qa$, and $Itgam$. Colocalization of IBA1 and TMEM119 in the cortex confirmed that the level of expression in individual microglia is unchanged in the $Il34^{-/-}$ brain (Fig 5C). Cluster 18 includes three transcription factor

---

mApple transgene. **(C)** Whole-mount imaging of the mesothelium (surface), and peritubular and interstitial regions of the testes. Tissues were directly imaged on an Olympus FV3000 confocal microscope with a 10x objective, and the scale bar represents 100 µM. **(D)** Half brains from $Il34^{+/+}$, $Il34^{+/-}$, and $Il34^{-/-}$ animals with the $Csf1r$-mApple reporter were homogenized into single-cell suspension and analysed by flow cytometry. Total cells were enumerated after extraction, and total numbers of microglia (CD45$^{low}$/Cd11b/c$^{high}$) and brain-associated macrophages (BAM; CD45$^{high}$/CD11b/c$^{low}$) were calculated from proportions derived from flow cytometry. Representative plots from each genotype are shown and quantified, and each dot represents an independent animal (n ≥ 12).

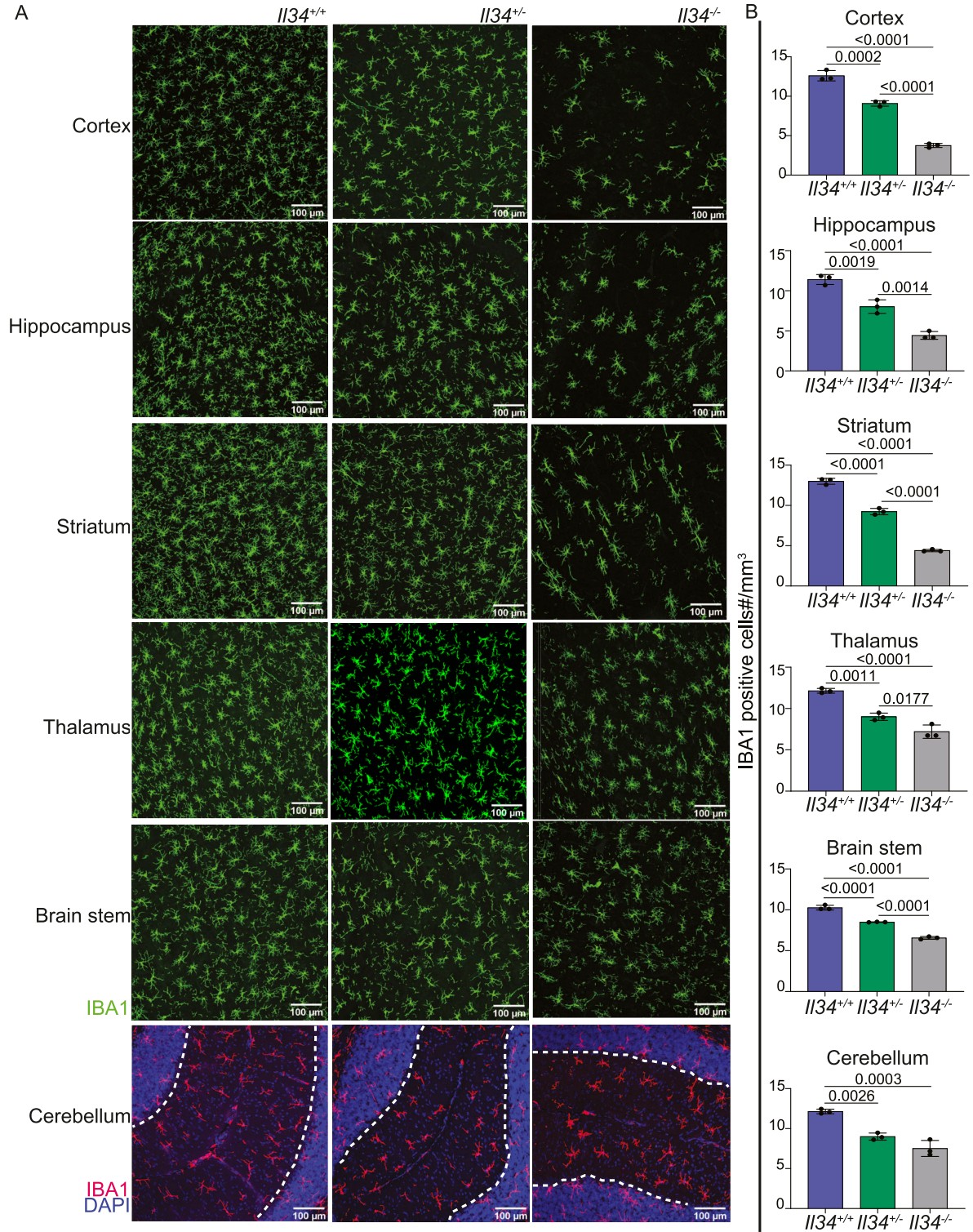

**Figure 3. Loss of IL34 leads to selective reduction of microglia in grey matter regions.**
**(A)** Representative images of brain sections stained for IBA1 (green or red) and DAPI (blue). Images were captured on the Olympus FV3000 confocal microscope. Grey matter in cerebellum is demarcated by a white dashed line. **(B)** Microglial density for each brain region (specifically grey matter in cerebellum) was quantified as described in the Materials and Methods section. Each point represents data from an individual animal. Statistical analysis was performed using a one-way ANOVA.

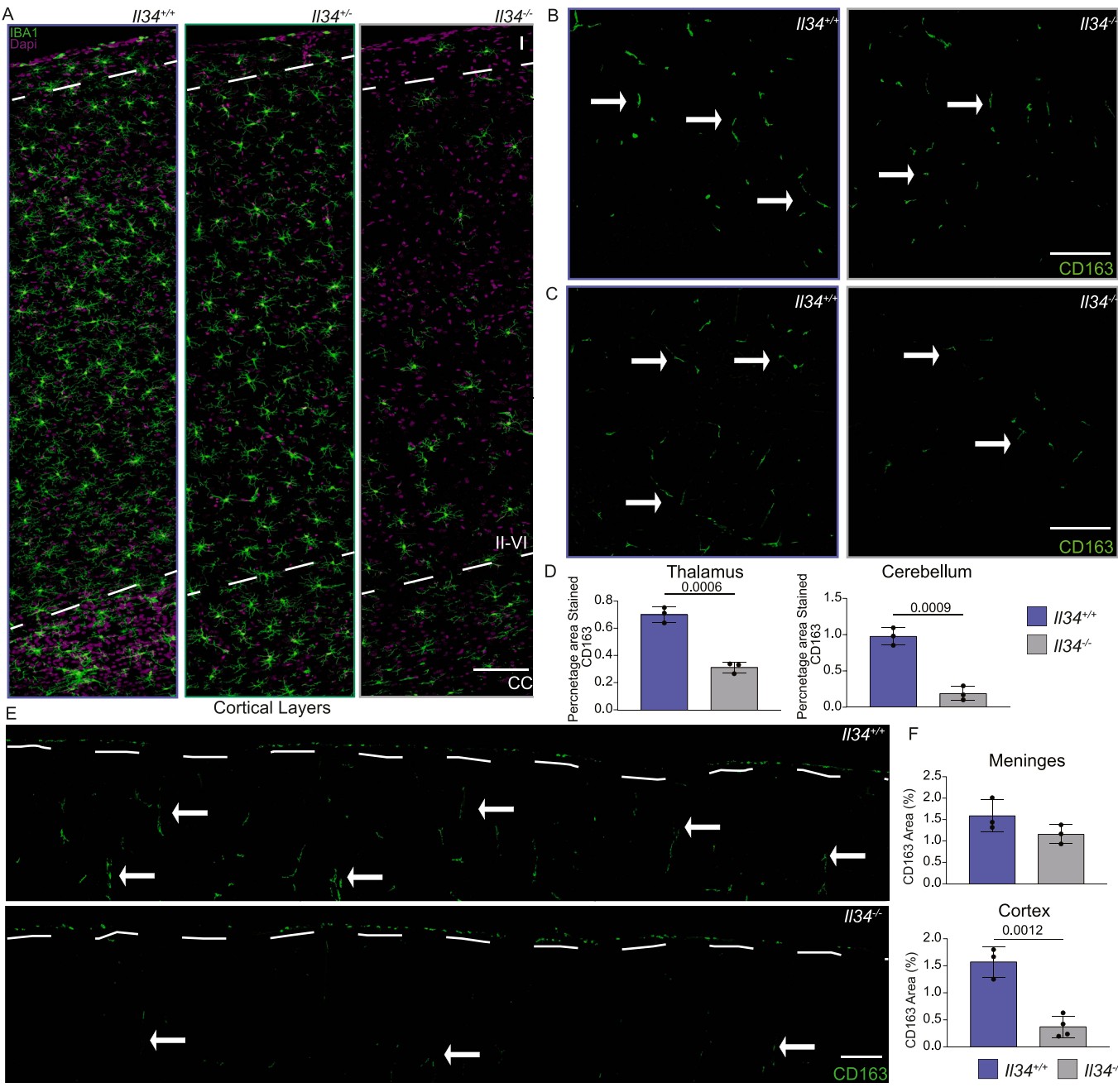

**Figure 4.  Gradient of microglial density and selective depletion of perivascular macrophages in grey matter of IL34-deficient rats.**
**(A)** Free-floating sections of cortex from *Il34*<sup></sup>*⁺/⁺*, *Il34⁺/⁻*, and *Il34⁻/⁻* brains stained with IBA1 (green) and DAPI (magenta). Panels show representative high-resolution scans of the cortical layers from the meninges and cortical layer 1 (at top), through layers II-VI to the corpus callosum (CC) at the bottom. Note the reduced density of IBA1⁺ cells in the CC (demarcated by white dotted line) and apparent gradient through the cortical layers. **(B, C, D, E, F)** CD163 (green) was used to identify perivascular macrophages. **(B, C, E)** Representative images of thalamus and cerebellum grey matter (GM) from WT and *Il34⁻/⁻* (B, C) and whole cortical scans (E) are shown. White dashed lines demarcate the meninges, and arrows indicate direction towards CC. Data are presented as percentage area stained with CD163 in different brain regions with each dot representing an independent animal and scale bars representing 100 μm. Statistical analysis was performed using a one-way ANOVA followed by Tukey's test.

genes, *Spi1, Irf5*, and *Irf8,* but excludes *Sall1*, which is required for microglial lineage identity ([68, 69]) ([Fig 5D]). Analysis of *Csf1r⁻/⁻* rat brains indicated that *Sall1* is not microglia-restricted ([66]). Markers of BAM are more difficult to detect in total RNA-seq data, but *Cd74* and *Cd68* were also reduced in *Il34⁻/⁻* brains. Separate sheets in Table S1 show transcripts ranked based on IL34KO/WT ratio with

known CSF1R-dependent microglia or macrophage-associated transcripts highlighted.

In overview, the data demonstrate that the transcriptomic profiles of IL34-independent (i.e., CSF1-dependent) microglial and BAM populations do not differ significantly from those of IL34-dependent populations located mainly in grey matter.

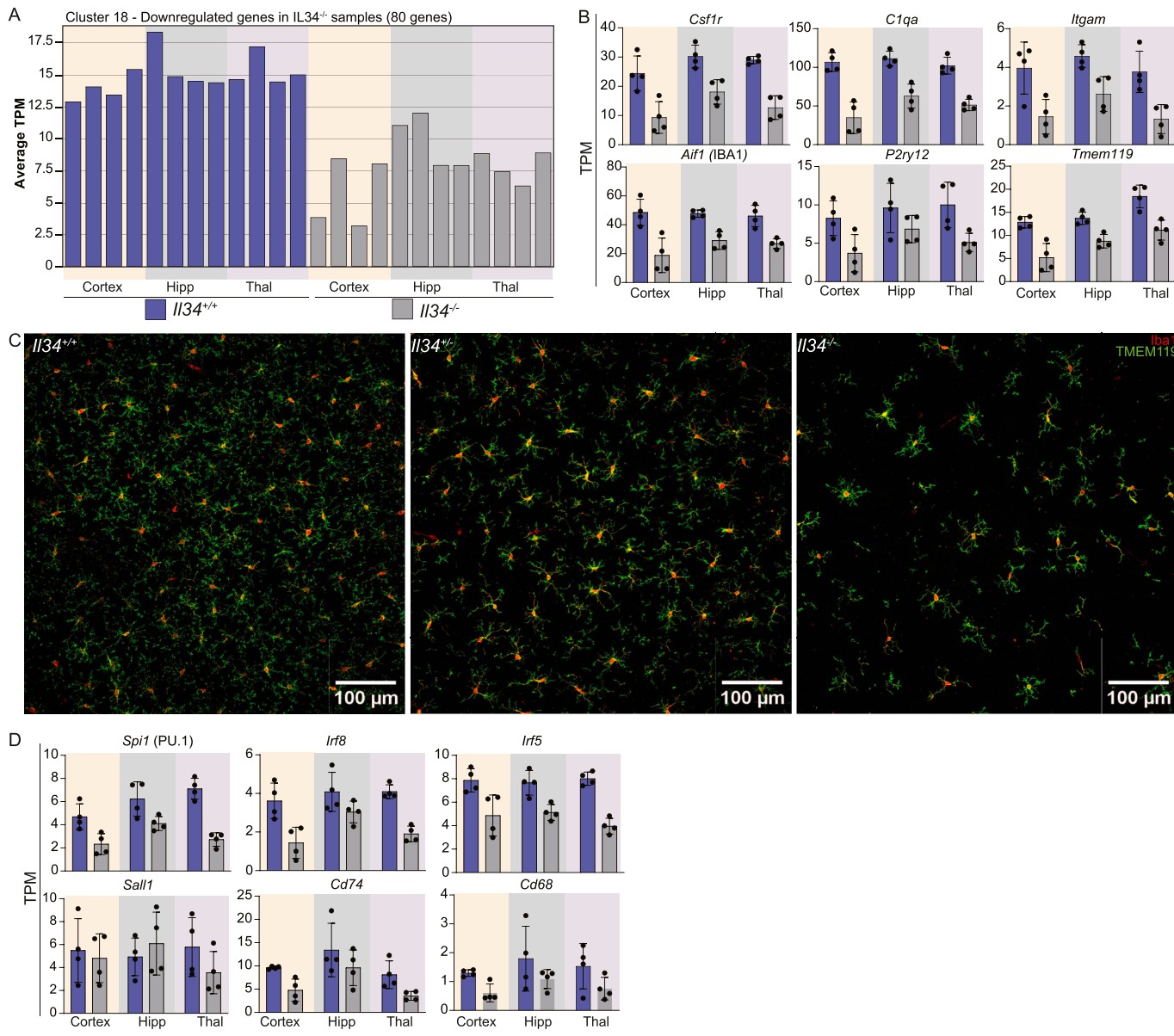

**Figure 5. Selective loss of the microglial gene signature in multiple brain regions in _Il34_$^{-/-}$ rats.**
mRNA was isolated from the cortex, hippocampus (hipp), and thalamus (thal) of 12-wk-old _Il34_$^{+/+}$ and _Il34_$^{-/-}$ rats (N = 4/genotype), and gene expression was quantified by RNA-seq as described in the Materials and Methods section. Network analysis to identify co-expressed gene clusters was performed using Graphia. The complete dataset and cluster analysis are provided in Table S1. Cluster 18 is the only cluster showing a relationship with the _Il34_ genotype. **(A)** Average tag per million count of all genes from Cluster 18 identified by network analysis is represented as a bar for each sample sequenced. **(B)** Each panel shows the expression of individual genes from Cluster 18; each dot represents an independent sample (n = 4). **(C)** Representative images of cortex from adult _Il34_$^{+/+}$, _Il34_$^{+/-}$, and _Il34_$^{-/-}$ brains (n = 3/genotype) stained with IBA1 (red) and TMEM119 (green). **(D)** Expression of genes encoding microglial transcription factors (_Pu.1, Irf5, Irf8_ and _Sall1_) and macrophage markers (_Cd74 and Cd68_).

## The effect of microglial deficiency on brain development and behaviour

Dominant or recessive _CSF1R_ mutations in humans are associated with microglial deficiency and consequent neuropathology ([70](navigation)). In contrast to _Csf1r_$^{-/-}$ rat brains, the _Il34_$^{-/-}$ brains showed no evidence of ventricular enlargement and the transcriptomic profiles did not include the induction of stress response genes (e.g., _Rbm3, Mt1/2_) reported in the brains of juvenile _Csf1r_$^{-/-}$ rats ([66](navigation)) ([Fig S3A](navigation)). In the

hippocampus of _Csf1r_$^{-/-}$ rats, we observed disorganized maturation of neuronal progenitors stained with the marker doublecortin (DCX) but considered that this might be a consequence of stress because of ventricular enlargement ([66](navigation)). Although microglia are depleted in the vicinity of the dentate gyrus in _Il34_$^{-/-}$ brains, DCX staining did not distinguish WT and mutant animals ([Figs 3B](navigation) and [S3B and C](navigation)). Consistent with this observation, the _Il34_ mutation had no significant effect on mRNA encoding markers of neurogenesis (_Dcx, Pax6, Neurod1, Nes_) in the hippocampus (Table S1). Furthermore, in

common with previous analysis of *Csf1r*$^{-/-}$ rat brains, the expression profiling did not reveal any effect of the loss of grey matter microglia on astrocytes. The expression of *Gfap* mRNA and the density, distribution, and morphology of GFAP$^+$ astrocytes were not affected by microglial deficiency (Fig S3D and E). In the cluster analysis, as a consequence of sample-to-sample variation in myelination, oligodendrocyte-specific transcripts (e.g., *Olig1, Mbp*) form part of a single cluster, Cluster 6, that is not correlated with the *Il34* genotype. Diffusion tensor imaging revealed no significant effect of the lack of IL34 on any white matter structure or the overall volume of any brain region (Fig S3F, Table S2). Notably, there were no ventricular enlargement and no detectable flares in the thalamus suggestive of calcification at 6 mo of age.

One advantage of the rat is the wide range of human-relevant behaviour and functional tests compared with mice (71). We subjected WT, *Il34*$^{+/-}$, and *Il34*$^{-/-}$ male and female rats to a battery of tests between 4 and 6 mo of age. We found no significant differences in performance in the sucrose preference (Fig S4A and D), O-maze (Fig S4B and E), and open field tests (Fig S4C and F).

To look for age-dependent pathology resembling human leukodystrophy or neurodegeneration, we analysed the same cohort at 18 mo of age. Quantification of IBA1$^+$ cells demonstrated that the selective loss of microglia in the cortex and hippocampus in *Il34*$^{-/-}$ and partial depletion in *Il34*$^{+/-}$ rats was still present at 18 mo, but although there appears to be a decrease in the thalamus, this was not significant (Fig S5A and B). Astrocytosis detectable by GFAP staining can be a sensitive indicator of neuropathology, accelerated in microglia-deficient mice (72, 73). However, the density and morphology of GFAP$^+$ astrocytes were not significantly different in any brain region in the aged animals (Fig S5C and D). In addition, we assessed [$^{18}$F]DPA714 uptake by PET/MR to measure neuroinflammation (74). This radioligand binds specifically to TSPO, a mitochondrial membrane protein, and is used to detect neuroinflammation in human patients (74, 75, 76, 77, 78, 79). We focussed our analysis on the hippocampus, thalamus, pituitary, and different cortex regions. We detected differences in uptake between male and female animals consistent with previous reports (80) and a male-specific increase in uptake with age. However, there was no detectable difference in uptake between genotypes (Table S3).

Recent studies have highlighted the relationship between microglial dyshomeostasis and the loss of perineuronal nets (PNN) associated with cortical parvalbumin interneurons (81, 82). Proteoglycans within these extracellular matrix structures can be detected by staining with *Wisteria floribunda agglutinin* (WFA), and we localized parvalbumin (PV) neurons in the cortex. We found no significant difference in WFA or PV staining between WT and IL34$^{-/-}$ rats (Fig S5E and F). Microglia deficiency in mice leads to reactive astrocytosis and extensive calcification in the thalamus detectable by 3–6 mo of age and progressing in severity (83, 84). Consistent with the lack of detectable astrocytosis in the thalamus of aged *Il34*$^{-/-}$ rats, there was no calcification detectable by labelled risedronate staining (Fig S5G). In summary, the only significant difference detected between WT and *Il34*$^{-/-}$ brains at any age examined is the selective loss of microglia in grey matter and associated reduction in detectable microglia-specific transcripts.

## The role of IL34 in chronic inflammation in the kidney

Network analysis of individual tissues with multiple region or state-specific RNA-seq data revealed a strong correlation between *Il34* mRNA and a macrophage-specific signature in the kidney (55), suggesting that IL34 does contribute to control of resident macrophage abundance in this organ. To determine the role of IL34 in chronic renal inflammation, we established the rat adenine diet model (85, 86), which leads to slowly progressive kidney damage with many features of chronic kidney disease in humans including tubular atrophy, interstitial fibrosis, and glomerulosclerosis. We exposed cohorts of equal numbers of male and female WT and *Il34*$^{-/-}$ rats to a 0.2% adenine diet or normal chow for 6 wk. Both WT and *Il34*$^{-/-}$ rats had normal growth when placed on the adenine diet (Fig 6A). At a cage level, increased water consumption was evident by 2 wk, and by 5 wk, it was more than doubled for both male and female cages (Fig 6B and C). As reported previously (86), the effect of the diet on renal function was sex-dependent. In males, we detected increased serum creatinine and blood urea nitrogen, whereas serum metabolites were not affected in females. These changes were not significantly reduced in the *Il34*$^{-/-}$ males (Fig 6D and E).

A feature of the pathology in this model is the deposition of adenine crystals in the cortex, which appeared reduced in females regardless of genotype, thickening of Bowman's capsule and mesangium in glomeruli, loss of tubular brush borders, tubular atrophy and dilatation, expansion of interstitial mononuclear cell populations, and extensive fibrosis detected with Sirius red (Figs 6F and G and S6A–C). Glomerular pathology appeared male-specific, perhaps associated with the observed sex difference in clearance function. Quantitative analysis did not reveal any significant effects of the *Il34* mutation, notably including the extensive interstitial fibrosis (Fig 6F and G).

We next asked whether the loss of IL34 had any effect on renal macrophage populations. Renal macrophages can be detected by staining for IBA1 and IBA1$^+$ cells are completely absent in the kidneys of *Csf1r*$^{-/-}$ rats (52). There was a massive expansion of IBA1$^+$ macrophage populations in the cortex and medulla of rats on the adenine diet (Fig 7A and B). Within the limits of quantification, the density of renal IBA1$^+$ cells in control animals was not significantly affected by the loss of IL34, but the response to the adenine diet was reduced by around 50% (Fig 7D and E). The presence of macrophages expressing the haptoglobin receptor, CD163, has been associated with kidney injury in both animal models and human disease and is commonly considered a marker of an anti-inflammatory phenotype (87). In the adenine model, we detected expansion of a CD163$^+$ cell population in the medulla only, which was significantly reduced in *Il34*$^{-/-}$ rats (Fig 7C and F).

To further analyse the effect of the IL34 knockout on renal pathology, we performed total RNA-seq analysis of kidneys from wild-type and *Il34*$^{-/-}$ male control and adenine-fed rats. The complete dataset is provided in Table S4. Separate Tabs show ranked lists of average expression comparing control versus adenine-fed for each genotype and WT versus *Il34*$^{-/-}$ for the adenine-fed and control animals. Table S4 also includes a highlights tab showing the data for selected adenine-regulated genes in different functional groups that provide markers of pathology and an indication of the impact of the

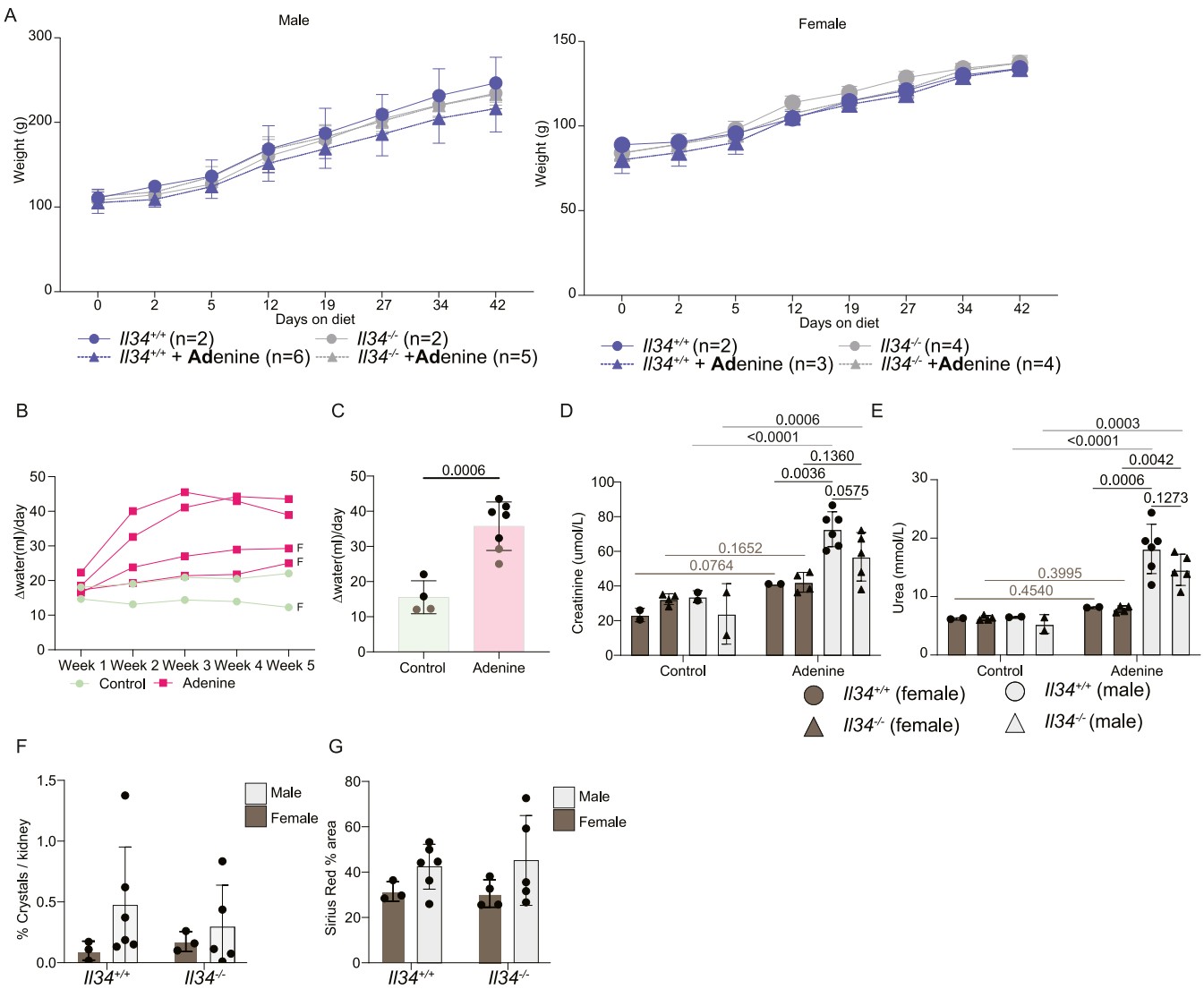

**Figure 6. Loss of IL34 does not affect pathology in the adenine diet–induced model of chronic kidney disease.**
Cohorts of male and female rats were fed normal chow or a diet containing 0.2% adenine for 6 wk. **(A)** Animal body weights were measured over the period of the adenine diet. **(B)** Water consumption by cage was quantified by comparing the difference in the weights of the bottle to the prior day; each line represents a single cage tracked longitudinally, and "F" denotes female cages. **(C)** Water consumption at the cage level for the 5th wk of the diet; each dot represents an independent cage: black dots are male cages, and brown dots are female cages (n ≥ 4). **(D, E)** Serum creatinine and urea were measured from serum collected through cardiac puncture. **(F)** Number of crystals in each kidney was quantified by calculating the area of the crystal across the entire kidney and normalized to the size of the kidney section quantified. **(G)** Sirius red staining was quantified as total stained area over the entire area of the kidney section quantified.

*Il34* mutation. *Csf1* and *Il34* mRNAs were each detectable in control kidney and increased >2-fold in response to the adenine diet. Consistent with the reduced detection of IBA1⁺ cells in situ, the detection of *Aif1* mRNA was increased ~6.5-fold in the kidneys of adenine-treated rats and this increase was reduced significantly in the *Il34⁻/⁻* rats. Network analysis of the complete dataset is also shown in Table S4. This analysis revealed a large cluster of 5,879 transcripts (Cluster 1) with a shared pattern of up-regulation in response to the adenine diet regardless of genotype, with a trend towards lower expression in the *Il34⁻/⁻* rats. We recently identified the kidney-resident macrophage signature by comparing the transcriptional profiles of kidney from wild-type and *Csf1r⁻/⁻* rats (56). 158/166 CSF1R-dependent transcripts identified by that analysis are

contained within Cluster 1 and are highlighted in red in Table S4. These transcripts were increased 4–40-fold in the kidney on the adenine diet and reduced in *Il34⁻/⁻* rats compared with WT (ratio 0.27–0.68; mean 0.46). The individual data for selected macrophage genes *Aif1*, *Adgre1*, *Csf1r*, *Clec10a*, *Mrc1*, and *C1qa* are shown in Fig S7A. One other feature evident in Cluster 1 is the transcriptional signature of the cell division cycle. The cluster contains 267 of 496 genes with known functions in the cell cycle (88) including *Mki67*. The overall pattern indicates that the response to adenine involves expansion of macrophage populations with a resident transcriptomic profile and that this expansion depends in part on induction of IL34. *Csf1* and *Il34* are not the only inducible transcripts encoding myeloid growth factors in the injured kidney. *Flt3lg* mRNA was detected at a similar

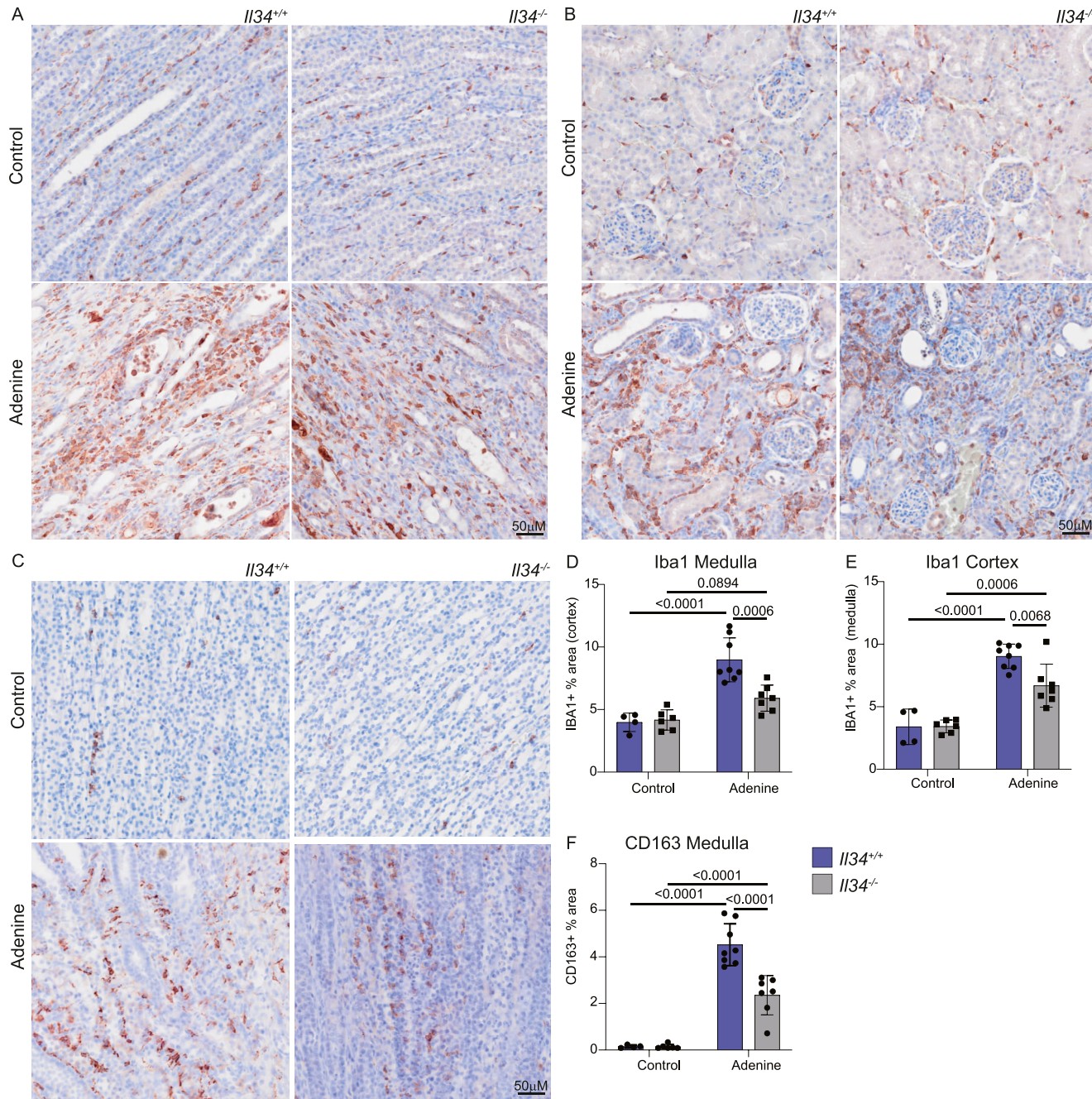

**Figure 7. IL34-dependent macrophage expansion during chronic kidney injury. Cohorts of male and female rats that were fed normal chow or a diet containing 0.2% adenine were stained for IBA1 and CD163.**
**(A, B, C)** Representative images for IBA1 in the medulla and cortex (A, B) and CD163 in the medulla (C) are shown. **(D, E)** Medulla and cortex from IBA1-stained kidneys were quantified as percentage of area stained. **(F)** CD163 was only detected and quantified in the medulla. Slides were scanned on an Olympus slide scanner (VS120 or VS200) with a 40x objective and images captured with OlyVIA software. Scale bars represent 50 µM.

level and increased in response to adenine, likely driving expansion of FLT3-expressing classical dendritic cells (cDC1) and increased detection of markers *Flt3, Ccr7, Xcr1, Itgae*, and *Clec9a* (89) (Fig S7B). Although macrophages appear to be the dominant inflammatory cell population, the RNA-seq data in Table S4 reveal numerous chemokines (e.g., *Ccl2, Cxcl1*), interferon-response genes (e.g., *Ifitm3, Ifit3*), and cell lineage–restricted signature genes associated with monocytes (*Cd14, Cybb*), T cells (*Cd3e/d/g*), B cells (*Cd19, Cd22, Ighm*), and granulocytes (*S100a8/9*) each increased in the injured kidney and to a lesser extent in the *Il34⁻/⁻* rats.

The tubular pathology and interstitial fibrosis observed in tissue sections are associated with markedly increased detection of markers of epithelial injury including cytokeratins (*Krt7/8/18/19*

(90)), *Havcr1* (aka Kidney Injury Marker 1), *Spp1* (osteopontin), and *Mmp7* (91) and of myofibroblasts (e.g., *Acta2, Col4a1, Mgp*). Cytokines *Il19* and *Il24*, implicated in epithelial injury and fibrosis in human kidney disease and in mouse models (92, 93), were almost undetectable in control and strongly expressed in diseased kidney. Consistent with evidence of inflammasome activation in crystal-induced nephropathy (94), *Il1b* and *Il18* were each increased. Epithelial loss or dysfunction is indicated by reduced detection of tubule markers such as *Umod, Slc12a1*, and *Slc21a4* defined in the rat transcriptomic atlas (55). Within the constraints of interindividual variation, there was no significant difference in detection of any of these markers of pathology between the wild-type and *Il34*$^{-/-}$ rats on the adenine diet (Table S4).

The histopathology and RNA-seq analysis indicate that IL34-dependent expansion of resident macrophages is not a driver of tubular damage or fibrosis. The alternative possibility is that it is a response to injury and associated with repair. To address this possibility, we looked at recovery from the adenine diet. WT and *Il34*$^{-/-}$ rats were fed the diet for 6 wk, then transferred to normal chow for 4 wk. After transfer to normal chow, both groups continued to grow normally. Serum creatinine and BUN trended downwards towards control levels but remained elevated independent of the genotype (Fig S8A and B). The expression of *Cd163* mRNA in WT animals returned to baseline and fibrosis scores based upon Sirius red staining also showed evidence of recovery, but there was no difference between WT and *Il34*$^{-/-}$ rats (Fig S8C and D). In summary, the adenine-induced renal pathology was slowly reversed after removal of the diet, but reversal was not altered by the absence of IL34.

## Discussion

A previous analysis of the phenotype of an *Il34* mutation in rats (57) was focussed on the putative expression of IL34 by regulatory T cells (Treg) and a role in autoimmunity (44, 45). The expression of *IL34* mRNA reported in rat and human T cells was detected by qRT–PCR and in humans by flow cytometry. The conclusion that IL34 is a T-cell lymphokine is not supported by other evidence. *IL34* mRNA is not detectable in human Treg in bulk (54, 95) or single-cell RNA-seq data (96) and was not detected in either CD4 or CD8 rat Treg isolated using a *Foxp3*-EGFP reporter by the same group (97). We were not able to reproduce the apparent loss of CD8 T cells reported in *Il34*$^{-/-}$ rats (57). Indeed, we found no detectable effect on leukocyte populations in the steady state. Similarly, no change in relative abundance of T-cell populations was detected in *Il34*$^{-/-}$ mice (24).

In mice, *Il34* mRNA is expressed selectively in epidermis and brain, whereas *Csf1* is expressed in mesenchymal cells and undetectable in epidermis (biogps.org). In contrast, in rats *Il34* mRNA is widely expressed. In the rat skin, *Il34* and *Csf1* mRNAs are expressed at similar levels (biogps.org/ratatlas) although the separation of dermis and epidermis has not been reported. In the mouse, Langerhans cells in the epidermis of the ear were almost entirely IL34-dependent (23, 24). Other squamous epithelia were not examined. In the *Il34*$^{-/-}$ rat, intraepithelial Langerhans-like cells

in squamous epithelia were not significantly depleted compared with WT (Fig 2A). Even in mice, CSF1 is required for repopulation of epidermal Langerhans cells after depletion (23). Our observations suggest that there is sufficient CSF1 available in rat skin to mitigate the impact of the loss of IL34 on Langerhans cell homeostasis.

The main location where the loss of IL34 impacted resident tissue MPS populations is the brain. A detailed analysis of microglial location and morphology in mice (98) indicated the relatively low density of F4/80$^+$ cells in white matter and fibre tracts, and the unique morphology aligned along the fibres. In common with humans (99), microglia in rat brain are equally abundant in white and grey matter. Consistent with the reported effect of anti-IL34 and anti-CSF1 treatment in adult mice (26), we observed selective loss of microglia in grey matter in *Il34*$^{-/-}$ rats, but the effect was subtle. Within the grey matter regions, CD163$^+$ perivascular macrophages were also depleted. In comparison with rats, the *Il34* mutation in mice had a greater overall impact on microglial density, including a reduced density in the corpus callosum (23, 24). Kana et al (27) reported the impact of conditional *Csf1* deletion in mouse brain, which produced selective microglial depletion in the cerebellum. Although *Csf1* is more highly expressed in the rat cerebellum than *Il34*, there was also selective loss of microglia within grey matter in this region and in the brain stem in the *Il34*$^{-/-}$ rats (Fig 3). This was not detected in *Il34*$^{-/-}$ mice (23, 24). Finally, we report for the first time that IL34 deficiency is not dosage-compensated and *Il34*$^{+/-}$ rats have a partial loss of grey matter microglia. In common with the reduced density of microglia in mice with a heterozygous dominant-negative *Csf1r* mutation (100), the residual microglia in the *Il34*$^{-/-}$ rat brain are evenly spaced. The regular distribution of macrophages in tissues has been attributed to competition for growth factors and mutual repulsion so that macrophages (including microglia) are autonomous and occupy a territory rather than a precise niche (2). The strict compartmentalization of IL34 and CSF1 implied from analysis of *Il34*$^{-/-}$ mice (23, 24) is clearly not the case in the rat. The gradient of microglial loss observed in the cortex (Fig 4A) suggests that CSF1 produced in corpus callosum is able to diffuse outwards to partly compensate for the loss of IL34 (101).

The expression profiling of the brain of *Il34*$^{-/-}$ rats revealed the coordinated partial loss of microglia-associated signature transcripts previously identified as CSF1R-dependent from analysis of multiple brain regions in *Csf1r*$^{-/-}$ rats (66). The thalamus was included because of recent evidence that the loss of microglia in mice leads to calcification of this brain region (83, 84). Aside from the loss of the microglial signature, we did not detect significant changes in the expression of any transcripts that would indicate either a neurodevelopmental defect or pathology. This observation provides evidence against a CSF1R-independent function of IL34 mediated through binding to PTPRZ1, which is expressed specifically by neurons (28). The absence of microglia in mice with a hypomorphic *Csf1r* mutation also leads to progressive demyelination, which has been analysed as a model for leukodystrophy associated with human *CSF1R* mutation (83, 102). Not surprisingly, because white matter microglia are largely spared, we did not detect changes in myelin in the *Il34*$^{-/-}$ rat. Although it is difficult to prove the absence of an effect, *Il34*$^{-/-}$ rats were not distinguishable from WT littermates in brain imaging and in a wide range of behavioural tests. Similarly, the density of perineuronal nets, a

morphological surrogate for neuronal plasticity in the cortex that is regulated by microglia in mice (81, 103), was not altered in $Il34^{-/-}$ rats. In overview, the analysis supports the view that microglia are not required for normal postnatal neuro-development (66, 104 *Preprint*). Their clear essential function in mice is in the mitigation of neuropathology (23, 24, 72, 73, 83, 102, 105). Interestingly, a relatively common stop-gain mutation in human *IL34* (Y213Ter, minor allele frequency ca. 0.1) has been associated with an increased relative risk of developing Alzheimer's disease (106). This variant lies downstream of the active IL34 peptide, so the functional impact on active IL34 generation is not known. IL34 amino acid sequence is highly conserved in mammals and birds (15), but available human exome sequences (https://gnomad.broadinstitute.org/) highlight multiple coding variants within the active peptide in human populations. The lack of dosage compensation and partial loss of microglia in $Il34^{+/-}$ rats indicate that heterozygous loss-of-function *IL34* mutation could impact microglia in humans. Equally, because homozygous mutants are viable, long-lived, and fertile, homozygous loss of function in humans may be tolerated. The $Il34^{-/-}$ rat provides a model in which to dissect the specific functions of microglia that more closely approximates human biology.

The emerging literature on IL34 views it as a pro-inflammatory cytokine and therefore a potential target in inflammatory disease (21, 42). Although there is clear evidence of an association between increased detection of IL34 and disease severity in a range of human inflammatory diseases, evidence of function is more equivocal and almost entirely based on mouse models. For example, Baek et al (46) reported that IL34 generated by tubular epithelial cells promoted macrophage-mediated acute epithelial injury in a kidney ischaemia/reperfusion model and exacerbated subsequent chronic fibrotic disease. This report is difficult to reconcile with the trophic effects of the alternative CSF1R ligand, CSF1, in the same model (50). Our comparative analysis of chronic renal injury in the WT and $Il34^{-/-}$ rats does not support the view that IL34 is an essential mediator of tissue injury. In WT rats in the adenine model, we detected induction of both *Il34* and *Csf1* mRNA, increased abundance of interstitial macrophages, and increased detection of numerous transcripts expressed specifically by resident kidney macrophage populations. Previous studies have demonstrated that the inducible expression of CSF1 drives local macrophage proliferation in response to renal injury in rats (48, 49). The relative density of IBA1+ cells, and detection of resident macrophage-associated transcripts, was approximately halved in the $Il34^{-/-}$ rats. The simplest explanation is that the two inducible CSF1R agonists each contribute to the response. FLT3 ligand may also contribute to expansion of interstitial mononuclear phagocyte populations. We observed the increased expression of *Flt3lg* mRNA and of transcripts associated with classical dendritic cells in response to the adenine diet (Table S4). There has been significant debate in the literature as to whether to classify renal interstitial myeloid populations as macrophages or DC based on surface markers (89, 107). Interestingly, expression profiling of kidneys of $Csf1r^{-/-}$ rats revealed 80% loss of *Flt3* mRNA, suggesting that renal interstitial myeloid cells may be unique in expressing both FLT3 and CSF1R (56).

Whereas the absence of IL34 compromised interstitial macrophage expansion in the injured kidney (Fig 7), it had no significant impact on any aspect of pathology, including clearance functions, overt pathology, or the expression of damage-associated transcripts (Table S4). Furthermore, the slow recovery after removal of the adenine diet was neither prevented nor accelerated. CSF1 and IL34 are also both induced after epithelial injury in the intestine, including in human ulcerative colitis (43). In both rats and mice, IL34 deficiency caused a moderate increase in pathology in the dextran sodium sulphate colitis model (43, 57, 108). Taking all of the evidence together, we suggest that the increased expression of CSF1R ligands is a consequence rather than a cause of pathology associated with chronic tissue injury that serves to promote resolution. That conclusion would imply that treatment with antagonists of either ligand or the receptor is unlikely to be therapeutic in inflammatory disease and may actually exacerbate or prolong the injury.

# Materials and Methods

### Sex as a biological variable

Both male and female animals were used unless otherwise stated.

### Animals

IL34 knockout rats were generated under contract by Taconic Biosciences through CRISPR-mediated recombination with guide RNAs flanking exon 2 of the *Il34* gene (Fig S1G). Animals carrying the knockout allele were imported to Australia, rederived into a specified pathogen-free facility, and maintained on the F344/DuCRL background. Deletion of exon 2 was confirmed by PCR using flanking primers (5'TTTAGCATCTCAGAGCCATCAG [forward], 5'GCAGCTTGACAACAGACAGG [reverse]) giving products of 1,541 bp (WT) and 353 bp (Mut). The same primers were used for genotyping.

There are no gross morphological differences in the IL34 homozygous or heterozygous animals, and both male and female animals are fertile. Rats between 8 and 10 wk of age were used to characterize distribution and morphology of macrophages unless stated otherwise. Rats for behavioural tests were 4 mo of age at the time of behavioural testing and 6 mo of age for PET/MR scanning. A cohort of animals were aged a minimum of 18 mo (18–26 mo) and also underwent PET/MR scanning. For whole-mount imaging, $Il34^{-/-}$ rats were crossed with the *Csf1r*-mApple transgenic reporter line (58). F1 progeny were interbred, and F2 animals 8–10 wk of age were used for analysis.

All animals were housed in ventilated cages at 21°C at a relative humidity of 76% and provided standard chow and water ad libitum. All animal procedures were performed at the University of Queensland under protocols approved by the Institutional Animal Ethics Unit (UQ AE2022/000676, AE2021/000337, AE2022/00066, and AE2022/000191).

### Adenine-induced chronic kidney injury model

Rats were 6 wk of age at the beginning of the adenine diet. Adenine-treated rats received R&M 0.2% Adenine Normal Feed (SF14-139; Specialty Feeds) and water ad libitum for 6 wk, whereas control

animals were kept on standard chow. The recovery cohort also received adenine for 6 wk and then were given standard chow for 4 wk.

## Animal behaviour studies

### Sucrose preference test

Animals were housed individually for 1 wk before testing. A two-bottle paradigm was set up for each cage with one bottle containing standard water and the second containing a 5% sucrose solution diluted in the standard drinking water. Bottles were weighed before testing and after removal. Animals began testing at 6 pm each evening, and bottles were replaced with standard water the next morning at 6 am and animals were tested again between 6 pm and 6 am with bottle positions swapped.

### Open field test

Animals were placed in the centre of a 56-cm square arena and allowed to explore for 10 min. A border 10 cm from the edge was considered the border boundary, and the remaining space within the square was considered the central area. Time spent within the central area was quantified when the centre of the animal's body occupied this space.

### Elevated O-maze

The platform is a circular disc ~120 cm in diameter. The platform width was ~10 cm and elevated 60 cm above the ground. The closed arms of the O-maze consisted of walls that were 30 cm tall. Animals were placed in the centre of the open arms and allowed to explore for 10 min. Total animal movement and time spent in the open arms were quantified.

### Micro-CT analysis of bone

Whole right hindlimbs were fixed in 4% PFA/PBS for 6 d at 4°C and then transferred into PBS and scanned using Bruker SkyScan 1272. All x-ray projections were reconstructed in a blinded manner using a modified back-projection reconstruction algorithm (NRecon 1.7.3.1 software (SkyScan; Bruker $\mu$CT)) to create cross-sectional images, and 3-dimensional (3D) reconstructions were viewed using CTvox 3.3.0 (Bruker $\mu$CT). Reconstructed images were then analysed using CTAn 1.19 software (Bruker), which has inherent 2D and 3D analysis tools (109).

### Flow cytometry and haematology

Blood was collected via cardiac puncture into EDTA-coated MiniCollect tubes (#450532; Greiner) for haematology or serum separator VACUETTE tubes (#456010; Greiner) for serum analysis. 100 $\mu$l of blood was treated with 1 ml of red blood cell lysis buffer (0.15 M NH4Cl, 10 mM NaHCO3) for 2 min at room temperature, then washed in 10 ml of PBS. Lysed preparations were then spun at 500$g$, 5 min, 4°C, washed in 10 ml of PBS, and then resuspended in FACS buffer. Thymus, cervical lymph nodes (cLN), spleen, and bone marrow were collected in FACS buffer (2% FCS/PBS) and filtered through a 40-$\mu$M filter to produce single-cell suspension. Splenic cell preparations were treated with red blood cell lysis buffer. All single-cell preparations were enumerated, and $1 \times 10^6$ cells were stained for 30 min at 4°C.

Brain tissue for flow cytometry was collected from one hemisphere by performing a sagittal incision down the midline of the brain. This tissue was then minced with a scalpel in a petri dish, resuspended in HBSS containing collagenase type IV (1 mg/ml), dispase (0.1 mg/ml), and DNase1 (19 mg/ml), and incubated on a rocking platform for 45 min at 37°C. The tissue was then passed through a 70-$\mu$m filter, washed in 25 ml PBS, and centrifuged (at 400$g$ and 4°C for 5 min). The supernatant was removed and the pellet resuspended in 12.5 ml isotonic Percoll (2.44 ml Percoll (Cytiva), 0.47 ml 10x PBS, 7.815 ml 1x PBS). The suspension was centrifuged at 800$g$ for 45 min (no brake) at 4°C. The supernatant containing the top myelin layer was aspirated, and the pellet was washed twice with PBS, and finally resuspended in flow cytometry (FC) buffer (PBS/2% FCS).

**The following antibody panel was used across tissue preparations:**

|  | Panel 1 | Panel 2 |
|---|---|---|
| Antibodies | CD32 | CD32 |
|  | HIS48-FITC | CD3-FITC |
|  | CD11B/C-BV510 | CD4-APC/Cy7 |
|  | CD45R-BV785 | CD8-APC |
|  | CD172-BV421 | CD45R-BV785 |
|  | CD45-PE/Cy7 | 7AAD |
|  | CD43-AF647 | |
|  | CD4-APC/Cy7 | |
|  | 7AAD | |
| Tissues profiled | Brain, bone marrow, spleen, and blood | Cervical lymph nodes and thymus |

Flow cytometry data were acquired on Fortessa X20 Cytometer (BD Biosciences), and data were analysed using FlowJo v10 Software (BD Life Sciences). Live single cells were gated and subsequent populations defined by the gating strategy illustrated in Fig S9A–D.

## Histochemical staining and immunolocalization

Spleens, livers, and kidneys were collected and washed in PBS before fixing in 4% PFA for 24 h at room temperature, then processed, and embedded into paraffin using routine methods. 4- and 10-$\mu$M sections were deparaffinized and rehydrated with standard methods: xylene followed by descending ethanol series. 4-$\mu$M sections were used for Sirius red and haematoxylin and eosin (H&E) staining and 10 $\mu$M sections for immunohistochemical staining as previously described (56, 110). Briefly, sections were stained with Gill's No. 2 haematoxylin for 30 s and eosin for 40 s for H&E staining, and for Sirius red staining, sections were stained in Picric–Sirius red solution (Australian Biostain P/L) for 1 h. Heat-induced epitope retrieval was performed in 10 mM sodium citrate buffer (pH 6.0) followed by staining with rabbit anti-IBA1 (FUJIFILM, 019-19741, 1:1,000; WAKO Chemicals). 10 mM EDTA buffer (pH 9.0) was used for antigen retrieval for rabbit anti-CD163 (ab182422, ERP19518, 1:500; Abcam) and CD209B (ab308457, 1:2,000; Abcam). Detection was performed with DAKO Envision anti-rabbit HRP detection reagent (Agilent). Sections were dehydrated in increasing ethanol series and cleared in xylene, and coverslips were mounted with DPX mountant (Sigma-Aldrich). Whole-slide digital scanning was performed on the VS120 or VS200 at 40x objective magnification.

Free-floating sections of brain were generated, and immunolocalization of markers was performed as described previously (66). For the older animals (>18 mo), tissues were fixed by transcardial perfusion with 4% paraformaldehyde in PBS and postfixed for 24 h at room temperature in 4% PFA/PBS. For analysis of young adult brains, sections were stained and imaged on an Olympus FV3000 microscope as confocal images on either the 40x or 60x oil objective with z-stacks using solid-state lasers (405, 488, and 647 nm) for each brain region (Fig S10A and B) as follows:

| Protein marker | Antibody details | Cell identification | Objective used | Stacks |
|---|---|---|---|---|
| IBA1 | Cat # 01-1874; Wako | Microglia | 40× air | 12 |
| GFAP | Cat # PA1-10004; Invitrogen | Astroglia | 40× air | 20 |
| DCX | 4604s; Cell Signalling | Immature neurons | 40× air and 60× oil | 20 and 30 |
| CD163 | ab182422; Abcam | Perivascular macrophages | 40× air | 15 |
| TMEM119 | 400-004; Synaptic System | Microglia | 20× air | 20 |

For analysis of aged animals (>18 mo), sections were stained and imaged on an Olympus FV3000 microscope as confocal images on either the 4x, 10x, or 20x oil objective with z-stacks using solid-state lasers (405, 488, and 647 nm) for each brain region as follows:

| Protein marker | Antibody details | Cell identification | Objective used | Stacks |
|---|---|---|---|---|
| IBA1 | Cat # 019-19741; Novachem | Microglia | 10x air | 30 |
| GFAP | Cat # PA1-10004; Invitrogen | Astroglia | 4x air | 40 |
| WFA | Cat # ZH0824; VectorLabs | Perineuronal nets | 20x air | 30 |
| PV | Cat # 195004; Synaptic Systems | Parvalbumin neurons | 20x air | 30 |
| Risedronate | Cat # BV500101; BioVinc LLC | Fluorescent bisphosphonate analogue | 4x air | 40 |

### Image analysis

Images were taken as a z-stack of 1 $\mu$m thickness and converted to z-projection on ImageJ with FIJI software (version 1.54f). A minimum of 4 images per brain section were analysed for either number of positive cells (IBA1, PV) or percentage area (GFAP, WFA) as follows: images were z-projected and converted to RGB image, and colour thresholding was performed using either default method (Iba1, GFAP) or MaxEntropy method (WFA, PV) with the same settings across genotypes. Raw data were collected using the measure tool, and values were plotted on GraphPad Prism (version 10.3.1).

### Diffusion tensor imaging and analysis

Fixed brains were washed in 0.1 M PBS for >1 wk for scanning using a 16.4 T vertical bore MR microimaging system (ParaVision v6.01; Bruker BioSpin), equipped with a Micro2.5 imaging gradient using a 20-mm linear surface acoustic wave coil (M2M; Brisbane). 3D T1/T2*-weighted multigradient echo images were acquired with TR = 70 ms, bandwidth = 100 KHz, FOV = 27 × 18 × 15 mm, and matrix size = 216 × 144 × 120, resulting in 120 $\mu$m 3D isotropic image resolution, with the acquisition time of 40 min. The multigradient echo data contained 15 TEs (=4, 8,12, .. 60 ms) for quantitative susceptibility mapping. Subsequently, samples were washed in 0.1 M PBS with 0.2% vol/vol Magnevist (Bayer) for >1 wk. 3D DWI data were acquired using a Stejskal–Tanner DWI spin-echo sequence with TR = 200 ms, TE = 21 ms, $\delta/\Delta$ = 2.5/12 ms, bandwidth = 50 KHz, FOV = 27 × 18 × 15 mm and matrix size = 180 × 120 × 100, image resolution = 150 $\mu$m, 30 direction diffusion encoding with b-value = 5,000 s/mm2, three b = 0 images, with the acquisition time of 15 h 42 min.

Quantitative susceptibility maps and T2* relaxation maps of the rat brains were calculated using the program (QSMxT) (111). The anatomical images were constructed by combining multi-echo images to enhance image contrast. Images were registered into the Waxholm Space MRI/DTI template for the Sprague Dawley rat brain, using ANTs diffeomorphic image registration (112, 113, 114). The model-based segmentation of brain regions was performed on each sample, and their image volumes and intensities were measured using ITK-SNAP.

The FID of DWI datasets was zero-filled by a factor of 1.5 in all dimensions before the Fourier transform to improve fibre tracking. DWI data were bias-corrected using ANTs N4BiasFieldCorrection and processed using MRtrix3 software (www.mrtrix.org). Fibre orientation distribution was reconstructed using the constrained spherical deconvolution method, and probabilistic tractography was performed using the iFOD2 algorithm. Tractography was performed for specific major white matter (WM) tracts. Firstly, the seeding regions of interest were manually drawn in the midsagittal and coronal sections of the colour vector map, and fibre tracts were generated for the corpus callosum (CC), hippocampal commissure (HC), internal capsule (IC), and anterior commissure (AC) at 100 seeds per voxel. The volumes of the white matter tracts were measured from Tract Density Intensity maps (TDI) with the intensity threshold set at 20% to remove background noise from probabilistic tracking (112, 115).

### PET/MR imaging of DPA714 uptake

DPA714-F18 was synthesized as previously described (78). Anaesthetized rats, with a cannulated tail vein (with a 0.020" ×0.060" OD Tygon microbore tubing, 45-cm-long and a 25 gauge needle), were placed in a combined MRI/PET system, comprising a Bruker AV4 70/30 BioSpec MRI including a 7T, 300 mm bore magnet with B-GA20S HP gradients and operating with ParaVision 360 V3.5. The PET images are acquired on a 3-ring PET insert with an axial FOV of 150 mm, transaxial FOV of 80 mm. A 40-mm ID rat head MRI rf coil inside the PET ring was

used to acquire head images simultaneously with the PET acquisition. A 60-min PET and dynamic GE images with 200 repetitions were started simultaneously. After ~2 min of scanning, the rats were injected with ~30 MBq of PET tracer solution, 300 $\mu l$ Gadovist, and saline/10% ethanol to give a total volume of 400 $\mu l$. This volume was injected manually via the catheter inserted into the tail vein in a slow bolus injection. After 12 min of dynamic GE scanning, the following MR scanning was performed during the PET acquisition:

PET reconstruction was performed using OSEM 0.5 with iterations = 1, Scatter correction, Randoms correction, Decay correction, Partial volume correction, point spread function correction, attenuation correction generated from the 3D FISP 250 $\mu m$ coronal sequence, and partitioning of 10 × 60 s and 10 × 300 s time frames. The dynamic PET series were then geometrically registered to the T2 turboRARE transverse image set and the 3D FISP 250 $\mu m$ image set. PET/MR data were analysed using PMOD software. PET and MR data for each animal were matched, registered onto the Schiffer rat brain atlas, and segmented (116). SUV/BW values were calculated after normalizing for decay and bodyweight of each animal scanned.

### Molecular profiling and RNA-seq of brain regions and kidney

Brain tissue from one hemisphere was minced and snap-frozen. RNA was extracted using TRIzol reagent using the standard manufacturer's protocol. 1 $\mu g$ of RNA was used to synthesize cDNA according to the manufacturer's protocol (Meridian Bioscience). qRT–PCR was performed with 2x SYBR Green reagent (Thermo Fisher Scientific) with the primer sequences (56). The cortex, hippocampus, and thalamus were sampled using a biopsy punch (1 mm diameter) and placed directly into TRIzol, and RNA was extracted. Whole kidneys were snap-frozen at collection, and RNA was extracted using TRIzol reagent.

Library preparation and sequencing were performed at the University of Queensland Sequencing Facility (University of Queensland, Brisbane, Australia). Bar-coded RNA-sequencing (RNA-seq) libraries were generated using the Illumina Stranded mRNA Library Prep Ligation kit (Illumina) and IDT for Illumina RNA UD Indexes (Illumina). The bar-coded RNA libraries were pooled in equimolar ratios before sequencing using the Illumina NovaSeq 6000 (Illumina). Paired-end 102-bp reads were generated using a NovaSeq S2 reagent kit v1.5 (200 cycles) (Illumina). After sequencing, fastq files were created using bcl2fastq2 (v2.20.0.422). Raw sequencing data (in the form of .fastq files) were provided by the sequencing facility for further analysis.

Raw reads were preprocessed using fastp v0.23.2 using previously described parameters (52, 117). FastQC was used on pre- and post-trimmed reads to ensure adequate sequence quality, GC content, and removal of adapter sequences (118).

The reference transcriptome used in this study was created by combining the unique protein-coding transcripts from the Ensembl and NCBI RefSeq databases of the Rnor6.0 annotation, as previously described (55). After preprocessing, the transcript expression level was quantified as transcripts per million using Kallisto (v0.46.0) (119). Kallisto expression files were imported into RStudio (R version 4.2.1) using the tximport package (v1.24.0) to collate

transcript-level TPM generated by Kallisto into gene-level TPMs for use by downstream tools.

All network cluster analysis was based on gene-level expression and was conducted using Graphia (https://graphia.app) (120). Only genes expressed at ≥ 1 TPM in at least two samples were retained for analysis. A network graph was then created where nodes (genes) were connected by edges (representing correlation coefficients). Relationships where r ≥ 0.85 were included. Samples that exhibited similar patterns of gene expression across tissues were placed close together in the network graph. Gene expression patterns were further characterized using the Markov Cluster Algorithm (MCL) with an inflation value of 2.0 to identify clusters of transcripts with similar expression patterns.

### Statistics

All data are presented as the mean ± SD and analysed with GraphPad Prism (9.0 or 10.0). Statistical testing was performed using an ANOVA with Tukey's post hoc analysis or t test for pairwise comparisons. $P$-values < 0.05 are considered significant. Statistical analysis for DPA-714-F18 uptake was undertaken using R version 4.4.1. Data were analysed using linear regression to assess the association between SUV-BW and genotype as the main predictor, sex and age cohort, and the interactions between these three variables where appropriate. Hierarchical model building was employed to determine the best-fit model for each region of the brain.

## Data Availability

All data are provided in supplemental file. Sequencing data can be accessed through accession number (PRJEB90514).

## Supplementary Information

## Acknowledgements

The authors would like to thank and acknowledge the team at the University of Queensland PACE Biological Resources Facility (BRF) for their assistance with husbandry, maintenance, care, and monitoring of animals. We would also like to thank and acknowledge the BRF staff at the QBI Animal Behavioural Facility and Centre for Advance Imaging for the monitoring, care, and maintenance of animals. The authors acknowledge the scientific and technical assistance from the microscopy, flow cytometry, and histology core facilities at Translational Research Institute and the National Imaging Facility, a National Collaborative Research Infrastructure Strategy (NCRIS) capability, at the Centre for Advanced Imaging, Australian Institute for Bioengineering and Nanotechnology, University of Queensland. We thank Glenda Gobe for advice on the renal pathology model. This work was supported by funding from Mater Foundation and NHMRC Investigator Grant #2007850 to DA Hume.

## Author Contributions

S Huang: conceptualization, data curation, formal analysis, supervision, validation, investigation, visualization, methodology, project administration, and writing—original draft, review, and editing.
OL Patkar: conceptualization, data curation, formal analysis, investigation, visualization, methodology, and writing—review and editing.
S Schulze: conceptualization, data curation, formal analysis, investigation, visualization, methodology, and writing—review and editing.
D Carter-Cusack: formal analysis, investigation, and visualization.
S Millard: formal analysis and investigation.
G Ranpura: formal analysis, investigation, visualization, and writing—review and editing.
EK Green: formal analysis and investigation.
E Maxwell: formal analysis and investigation.
J Kanesarajah: formal analysis and investigation.
G Cowin: data curation, investigation, and methodology.
D Stimson: resources.
ND Kurniawan: investigation, methodology, and writing—review and editing.
S Keshvari: formal analysis and investigation.
R Allavena: formal analysis, investigation, and writing—review and editing.
AR Pettit: supervision.
KM Irvine: conceptualization, supervision, funding acquisition, and writing—original draft, review, and editing.
DA Hume: conceptualization, supervision, funding acquisition, and writing—original draft, review, and editing.

## Conflict of Interest Statement

The authors declare that they have no conflict of interest.

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
