## [Reviewer comments · Life Science Alliance]

Life Science Alliance

Mutation in the rat interleukin 34 gene impacts macrophage development, homeostasis and inflammation

Stephen Huang, Omkar Patkar, Sarah Schulze, Dylan Carter-Cusack, Susan Millard, Ginell Ranpura, Emma Green, Emma Maxwell, Jeeva Kanesarajah, Gary Cowin, Damion Stimson, Nyoman Kurniawan, Sahar Keshvari, Rachel Allavena, Allison Pettit, Katharine Irvine, and David Hume

DOI: <https://doi.org/10.26508/lsa.202503264>

Corresponding author(s): David Hume, University of Queensland and Katharine Irvine, The University of Queensland

Review Timeline:

Submission Date:	2025-02-10
Editorial Decision:	2025-03-17
Revision Received:	2025-04-28
Editorial Decision:	2025-05-22
Revision Received:	2025-06-05
Accepted:	2025-06-05

Scientific Editor: Tim Fessenden

Transaction Report:

March 17, 2025

Re: Life Science Alliance manuscript #LSA-2025-03264-T

David A Hume
University of Queensland
Macrophage Biology Research Group
Brisbane, Qld 4102
Australia

Dear Dr. Hume,

Thank you for submitting your manuscript entitled "Mutation in the rat interleukin 34 gene impacts macrophage development, homeostasis and inflammation" to Life Science Alliance. The manuscript was assessed by expert reviewers, whose comments are appended to this letter. We invite you to submit a revised manuscript addressing the Reviewer comments.

Thank you for this interesting contribution to Life Science Alliance. We are looking forward to receiving your revised manuscript.

Sincerely,

B. MANUSCRIPT ORGANIZATION AND FORMATTING:

Reviewer #1 (Comments to the Authors (Required)):

With this manuscript, Huang and colleagues describe the impact of Il34 deletion in rats, focusing on various TRM populations, including microglia, Langerhans cells, and kidney macrophages. They show that IL34 deficiency selectively reduces gray matter microglia without causing major neurological phenotypes or behavioral deficits. Interestingly, perivascular macrophages also appear partially depleted in KO rats. Unlike in mice, IL34-deficient rats exhibit reduced microglia in the cerebellum, highlighting species-specific differences in IL34 expression patterns. Additionally, the study demonstrates that while IL34 contributes to macrophage expansion in chronic kidney inflammation, its deletion does not affect overall disease progression or fibrosis. Thus, the authors conclude that IL34 may not be an effective therapeutic target for inflammatory kidney diseases.

Overall, the manuscript is well-prepared, and the data are clearly presented. Although some differences exist between mice and rats (e.g., the broader Il34 expression pattern in rats), these findings align with previous studies, confirming that IL34 is crucial for the survival of microglia (and likely kidney macrophages) across species. The data presented here are relevant to the journal's scope and will be of interest for the neuroimmunology community. This reviewer looks forward to seeing an EAE model in these rats. Please see our minor comments below:

- 1) In their RNA-seq analysis of kidneys from adenine-fed rats, the authors suggest that kidney inflammation involves the recruitment of monocytes, T cells, and granulocytes, which is reduced in Il34 KO rats. Can the authors validate these findings by flow cytometry?
- 2) The authors report no difference in microglia and BAM numbers between HET and KO rats by flow cytometry, yet IF quantification shows significant differences. This discrepancy is confusing. Did the authors use counting beads or normalize absolute cell numbers to tissue weight? If not, quantification from brain section imaging may be more reliable. If microglia depletion in KOs is greater than in Hets, the sentence in line 181 should be rephrased.
- 3) The term "TSS" is introduced in line 125 but first used in line 117.

Reviewer #2 (Comments to the Authors (Required)):

I congratulate the authors for this well executed study with clear data and a concise reporting. My only point is about the last sentence in the abstract where they concluded that "We suggest that IL34 and CSF1 provide redundant signals to sustain microglia and to direct macrophage recruitment and repair tissue injury in the periphery." that appears in contradiction with the observations that IL34 and CSF1 regulates specifically the differentiation of microglia in unique regions while loss of IL34 loss affects only Langerhans cells etc... this does not appear redundant but quite specific....Can the authors clarify what they want to say and modulate such conclusion.

Reviewer #3 (Comments to the Authors (Required)):

The manuscript provides a thorough characterization of the effects of IL34 deficiency on phagocytes in the rat. In addition, the authors examine the impact of reduction of IL-34-dependent macrophage populations on brain anatomy and function and in peripheral inflammation. Overall, this is an interesting paper that provides useful information for scientists interested in macrophage biology. Several conclusions seem premature and inclusion of additional information and/or increasing sample size is recommended to sustain them. The author's attention is drawn to the following points:

Major:

- 1) Quantitative data should be provided for or Figs 1D, E and H, Fig 2A, B and supplementary fig. 1E.
- 2) Lines 183-195 and Fig3: the description of morphological changes in microglia should be accompanied by morphometric data documenting the decrease in process length and/or ramification. In the absence of counterstaining, is hard to evaluate the alignment of microglia with the fibre tracts. DAPI signal should be included in merged images as shown for the Iba1-stained cerebellum.
- 3) Fig 4: counterstaining should be shown in panels B, C and E. The notion of an "outward gradient of microglia and

perivascular macrophages" (lines 194-195) should be better explained. While indeed the density of cortical microglia decreases proportionally with the distance from the white matter, in the absence of counterstaining the gradient of perivascular macrophages is less obvious. The position of the corpus callosum should be indicated in panel E. In addition, a diagram showing the position of the areas examined in B and C within the thalamus and cerebellum is needed.

4) Fig. S3 B, C: the conclusion that IL34 deficiency does not affect the density of DCX+ cells in the hippocampus and therefore hippocampal neurogenesis might be premature. There is high variability among wt samples and, at n=3, the statistics might be misleading. A larger sample size is necessary to firmly support the conclusion.

5) Fig. S3D: please indicate the position of the area examined in the cortex and provide the counterstaining image.

6) Fig. S4C, F: may rats seem to spend very little time, if any, exploring either object and would typically be excluded from analysis due to lack of exploratory activity. The test should be repeated with validated objects that prompt exploration. Rats exploring for less than 5 sec total time should be excluded from the analysis.

7) Fig. S5 A, B: the decrease in Iba1+ cells does not seem to be significant in the thalamus. Pls correct statement in the main text, line 249.

8) Fig. S5, C, D: again a conclusion that might be premature at least for the cortex. An increase in sample size is necessary to sustain the conclusion in lines 251-253 of the main text

9) Lines 286-289 and fig 6D, E: p values for comparisons between control and adenine diets should be shown to sustain the conclusions in the main text.

10) Fig S8A, B: p values for comparisons between control and recovery should be provided. A color legend should be added.

Minor:

1) Line 117: define TSS here instead of line 126

2) Line 287: in animals, the effect is sex, not gender dependent.

We thank all the reviewers for their time and appreciate the comments and feedback.

Reviewer #1:

1) In their RNA-seq analysis of kidneys from adenine-fed rats, the authors suggest that kidney inflammation involves the recruitment of monocytes, T cells, and granulocytes, which is reduced in Il34 KO rats. Can the authors validate these findings by flow cytometry?

The quantitation of mRNA encoding sets of lineage-specific genes (cell-type signatures) within total RNA-seq data is widely used as a way of assessing changes in cell composition. One example is Imsig (PMID: 30266715), which has been cited 200 times. We have altered the text to be more precise. We do not believe that Flow Cytometry would add additional weight to our conclusions and additional experiments performed solely to address this comment would be difficult to justify on animal ethics grounds.

2) The authors report no difference in microglia and BAM numbers between HET and KO rats by flow cytometry, yet IF quantification shows significant differences. This discrepancy is confusing. Did the authors use counting beads or normalize absolute cell numbers to tissue weight? If not, quantification from brain section imaging may be more reliable. If microglia depletion in KOs is greater than in Hets, the sentence in line 181 should be rephrased.

The line in 181 and related statements have been rephrased (line 191/192). There are no differences in the brain weight or total body weight of wildtype, heterozygous or knockout animals (data below). The Flow Cytometry data has a greater variance likely arising from differences in recovery of microglia following disaggregation. Despite this variance, the reduction in microglia in heterozygotes relative to WT is significant.

[Figure removed by editorial staff per authors' request]

3) The term "TSS" is introduced in line 125 but first used in line 117.

TSS is now defined in line 117.

Reviewer #2:

"We suggest that IL34 and CSF1 provide redundant signals to sustain microglia and to direct macrophage recruitment and repair tissue injury in the periphery." that appears in contradiction with the observations that IL34 and CSF1 regulates specifically the differentiation of microglia in unique regions while loss of IL34 loss affects only Langerhans cells etc... this does not appear redundant but quite specific....Can the authors clarify what they want to say and modulate such conclusion.

We agree this was confusing and have reworded to be more concise. The redundancy lies in the fact that the two ligands can each activate via CSF1R to produce the same outcome (proliferation/survival). The RNA-seq data indicates that IL34-dependent microglia do not have a unique expression profile; the only effect of loss of IL34 is to reduce microglial density.

Reviewer #3:

Major:

1) *Quantitative data should be provided for or Figs 1D, E and H, Fig 2A, B and supplementary fig. 1E.*

In the CSF1R knockout rat there is a complete loss of marginal zone macrophages at the molecular level and by histological analysis. In the representative images we are demonstrating that these cells are not lost, we used CD209b to highlight this.

[Figure removed by editorial staff per authors' request]

Our intention in showing images was not to quantify the relative loss of Langerhans cells (LC) but to emphasise that by contrast to mouse IL34KO, the rats are not absolutely LC-deficient. In response to the reviewers comment we have also quantified LC in multiple replicates of the tissues shown in Figure 2A (Data in Figure S2A). The outcome makes the point more strongly in that the loss of LC in the skin is not significant. We have modified the discussion to make this point more strongly, also citing data from the mouse indicating that CSF1 can contribute to LC repopulation.

2) Lines 183-195 and Fig3: the description of morphological changes in microglia should be accompanied by morphometric data documenting the decrease in process length and/or ramification. In the absence of counterstaining, is hard to evaluate the alignment of microglia with the fibre tracts. DAPI signal should be included in merged images as shown for the Iba1-stained cerebellum.

The main conclusion from this figure is that microglia are reduced in an IL34 dose-dependent manner. We agree that the comments related to morphology are not sufficiently supported. We have removed the lines in the results and discussion relating to the morphology of the residual microglia.

3) Fig 4: counterstaining should be shown in panels B, C and E. The notion of an "outward gradient of microglia and perivascular macrophages" (lines 194-195) should be better explained. While indeed the density of cortical microglia decreases proportionally with the distance from the white matter, in the absence of counterstaining the gradient of perivascular macrophages is less obvious. The position of the corpus callosum should be indicated in panel E. In addition, a diagram showing the position of the areas examined in B and C within the thalamus and cerebellum is needed.

The apparent loss of perivascular macrophages is inferred from Figure 4A as they are also IBA1 positive. Figure 4E&F demonstrates the loss of these macrophages within the outer layers of the cortex and not at the meninges, indicated with a white dashed line. Arrows have been included to highlight CD163+ macrophages in Figure 4B-E and scanning maps of these panels are now included in Supplemental Figure S10B.

4) Fig. S3 B, C: the conclusion that IL34 deficiency does not affect the density of DCX+ cells in the hippocampus and therefore hippocampal neurogenesis might be premature. There is high variability among wt samples and, at n=3, the statistics might be misleading. A larger sample size is necessary to firmly support the conclusion.

We have changed the statement in the figure legend title. The lack of effect of the IL34KO on DCX+ cell density in the hippocampus is supported by the RNA seq data for the hippocampus, which show that the IL34KO has no significant effect on *Dcx* mRNA, or on cell cycle-associated transcripts (Supplemental Fig3A). This point is now made more explicitly in the text.

5) Fig. S3D: please indicate the position of the area examined in the cortex and provide the counterstaining image.

The region of the Motor cortex was examined and a schematic of the area scanned is provided in supplemental Figure S10A).

6) Fig. S4C, F: may rats seem to spend very little time, if any, exploring either object and would typically be excluded from analysis due to lack of exploratory activity. The test should be repeated with validated objects that prompt exploration. Rats exploring for less than 5 sec total time should be excluded from the analysis.

We agree with this assessment of the novel object recognition test and have removed panels and statements relating to this assay.

7) Fig. S5 A, B: the decrease in Iba1+ cells does not seem to be significant in the thalamus. Pls correct statement in the main text, line 249.

This line has been corrected.

8) Fig. S5, C, D: again a conclusion that might be premature at least for the cortex. An increase in sample size is necessary to sustain the conclusion in lines 251-253 of the main text

The statement has been changed.

9) Lines 286-289 and fig 6D, E: p values for comparisons between control and adenine diets should be shown to sustain the conclusions in the main text.

P values between control and adenine have been added.

10) Fig S8A, B: p values for comparisons between control and recovery should be provided. A color legend should be added.

P values and legend have been added.

Minor:

1) Line 117: define TSS here instead of line 126

TSS is now defined in line 117

2) Line 287: in animals, the effect is sex, not gender dependent.

Line 287 has been changed.

May 22, 2025

RE: Life Science Alliance Manuscript #LSA-2025-03264-TR

Prof. David A Hume
University of Queensland
Macrophage Biology Research Group
-
Brisbane, Qld 4102
Australia

Dear Dr. Hume,

Thank you for submitting your revised manuscript entitled "Mutation in the rat interleukin 34 gene impacts macrophage development, homeostasis and inflammation". This manuscript was returned to original reviewers 1 and 3, whose reports are appended below.

As you will see Reviewer 1 recommends publication of this work. Like this reviewer, we were alarmed by the unnecessary and unjustified statement that flow cytometry staining would not strengthen claims made from transcriptomics. That said, ethical guidelines limiting animal experiments are perfectly understandable, and the adjusted text related to this point is satisfactory. Reviewer 2 noted multiple important issues on validation and discordant data and claims that must be resolved prior to publication. With the exception of point 2 from this reviewer, all points must be addressed in a revised manuscript which will be evaluated without further reviewer input. Finally, because LSA does not permit claims that are not supported by data ("Data not shown"), please remove the statement on risedronate staining in lines 268-270.

We would be happy to publish your paper in Life Science Alliance pending the above required revisions and final changes necessary to meet our formatting guidelines.

- Please add ORCID ID for corresponding (and secondary corresponding) author--you should have received instructions on how to do so.
- Please add the X and Bluesky handles of your host institute/organization as well as your own or/and one of the authors in our system.
- Please consult our manuscript preparation guidelines <https://www.life-science-alliance.org/manuscript-prep> and make sure your manuscript sections are in the correct order.
- Please add callouts for Figures S1G; S4A-F; S6A-C; S9A-D and S10A-B to your main manuscript text.

A. FINAL FILES:

B. MANUSCRIPT ORGANIZATION AND FORMATTING:

Sincerely,

Reviewer #1 (Comments to the Authors (Required)):

The authors have introduced minor text revisions without performing the suggested experiments. The assertion that RNAseq itself is sufficient to assess the immune infiltration into a tissue is nonsensical. The authors rejected the comments on the ground that the ethics committee in their country won't approve the euthanasia of additional animals for validation experiments. Although not entirely satisfied by the author's responses, this reviewer feels that the manuscript remains valid overall, and these findings are interesting for the field. This study is suitable for publication on LSA.

Reviewer #3 (Comments to the Authors (Required)):

The manuscript has been improved however, there are still some inaccurate statements and conclusions that need to be corrected:

1. The results of LC quantification shown in Fig S2A should be included in the main Fig. 2. This way it will become apparent that, even with the low sample size (n=2) provided, there is a significant decrease in the epithelium of the oesophagus although that is now denied in text (lines 159-162). The statement in the main text should be revised to reflect the conclusion provided in the rebuttal i.e. "by contrast to mouse IL34KO, the rats are not absolutely LC-deficient".
2. Given that morphometric analysis was not performed, the text in lines 176-181 should be revised by removing references to the ramified appearance of microglia. As it stands now, it implies that ramified microglia are selectively depleted.
3. It is unclear why the authors refuse to provide images containing the DAPI counterstaining and choose to provide a general map instead. This is necessary to determine whether the fields shown in each panel come from the comparable anatomic areas

of the brain. Please include the counterstaining in Fig 4 B, C, E and in Fig. S3B, D.

4. Given that the only test of cognitive performance was removed, the data shown in Fig. S4 are now insufficient to claim that absence of IL34 is not associated with behavioral alterations. This part should be removed from the abstract and the description of the novel object recognition test should be removed from the methods section.

5. In contrast to Fig 6 D, E, figure S8A shows that the 6 week adenine diet does not increase creatinine or urea levels in male IL34^{-/-} rats. Creatinine: IL34KO male untreated vs treated $p=0.0006$ in Fig 6D ($n= 2$ vs $n=5$) while in Suppl. Fig.8A, $p=0.071$ ($n=6$ vs $n=9$). Similarly, urea levels in IL-34KO rats are unchanged in the experiment involving more animals/condition : $p= 0.0938$ in Suppl. Fig.8B versus $p=0.0003$ in Fig 6E. How do the authors reconcile these discrepant findings? What is their final conclusion regarding the effect of IL-34 on kidney inflammation?

Dear Tim,

Thank you for your review and consideration of our manuscript for publication titled:

Mutation in the rat interleukin 34 gene impacts macrophage development, homeostasis and inflammation

Please find enclosed our response to reviewer #3 and relevant updates to the manuscript.

We have also addressed the 'data not shown' statement by including the risedronate data as Supplemental Figure S5G.

We look forward to the outcome of your consideration.

Yours sincerely

Professor David Hume AO, FMedSci, FRSE

Reviewer #3 (Comments to the Authors (Required)):

The manuscript has been improved however, there are still some inaccurate statements and conclusions that need to be corrected:

1. The results of LC quantification shown in Fig S2A should be included in the main Fig. 2. This way it will become apparent that, even with the low sample size (n=2) provided, there is a significant decrease in the epithelium of the oesophagus although that is now denied in text (lines 159-162). The statement in the main text should be revised to reflect the conclusion provided in the rebuttal i.e. "by contrast to mouse IL34KO, the rats are not absolutely LC-deficient".

The statement in the rebuttal was added to the main text in the first resubmission (now highlighted). Figure 2 now contains the quantification and lines in 158 have been altered to highlight the observation in the oesophagus.

2. Given that morphometric analysis was not performed, the text in lines 176-181 should be revised by removing references to the ramified appearance of microglia. As it stands now, it implies that ramified microglia are selectively depleted.

The word ramified has been removed.

3. It is unclear why the authors refuse to provide images containing the DAPI counterstaining and choose to provide a general map instead. This is necessary to determine whether the fields shown in each panel come from the comparable anatomic areas of the brain. Please include the counterstaining in Fig 4 B, C, E and in Fig. S3B, D.

We included DAPI in some images for Figure 3A, 4A and S2A because they do provide context at this magnification for example to distinguish grey and white matter in the cerebellum.

As indicated in the original submission and stated in the methods and the legends to figures the low power imaging of DAPI was used specifically to ensure that the fields that were analysed were indeed taken from comparable anatomic areas of the brain. This would not be evident from a higher power image. We have updated supplemental figure 10 to show all the brains imaged for Figure 4B-E.

We did not take higher power images of DAPI staining from the original material and are therefore unable to add counter staining to the higher power images as requested. This would require an additional set of experiments. We do not believe that the addition of the counter staining is required to support the conclusions.

4. Given that the only test of cognitive performance was removed, the data shown in Fig. S4 are now insufficient to claim that absence of IL34 is not associated with behavioral alterations. This part should be removed from the abstract and the

description of the novel object recognition test should be removed from the methods section.

Statements relating to behavioural alterations have been removed in the abstract and methods for novel object recognition has been removed.

5. In contrast to Fig 6 D, E, figure S8A shows that the 6 week adenine diet does not increase creatinine or urea levels in male *Il34*^{-/-} rats. Creatinine: *Il34*KO male untreated vs treated $p=0.0006$ in Fig 6D ($n=2$ vs $n=5$) while in Suppl. Fig.8A, $p=0.071$ ($n=6$ vs $n=9$). Similarly, urea levels in *Il-34*KO rats are unchanged in the experiment involving more animals/condition : $p=0.0938$ in Suppl. Fig.8B versus $p=0.0003$ in Fig 6E. How do the authors reconcile these discrepant findings?

We believe the presumed discrepancy occurs from an ambiguity in looking at data from males only (which have worse pathology) and mixed groups. We have clarified the text.

The control and adenine data shown in Figure S8A-C contains all animals (male and female) from Figure 6. Figure 7 also contains data from both males and females which illustrates the increase in macrophages in the kidney when fed adenine. The increase in IBA1 and CD163 in the adenine treatment is the same in both male and female animals (Figure 7D-F). The pathology measured by serum urea and creatinine, renal pathology by H&E and presence of adenine crystals is more severe in males compared to females. Hence our statement: *'the effect of diet on renal function was sex-dependent'*. We have not suggested the increase in macrophages is altering the function of the kidneys in fact we suggest the increase in macrophages is a consequence of injury.

We included the data in Supplemental Figure 8A-C to illustrate the direction of the effect of the recovery cohort. The inclusion of the female animals in the graphs hides the effect of the recovery cohort. However, the direction of the effect is still obvious.

The recovery animals contains only male animals because only male animals had altered kidney function on the diet i.e. Figure 6. Supplemental Figure 8 shows that after cessation of the diet there is no difference between wildtype and *Il34*^{-/-} in kidney pathology and function.

What is their final conclusion regarding the effect of *Il-34* on kidney inflammation?

Ultimately, our conclusion is unchanged. That is the loss of *Il34* does not exacerbate or ameliorate kidney inflammation which is likely due to overlapping function with *CSF1* which is also present in the inflamed kidney.

*'Our comparative analysis of chronic renal injury in the wild-type and *Il34*^{-/-} rats does not support the view that *Il34* is an essential mediator of tissue injury'*

June 5, 2025

RE: Life Science Alliance Manuscript #LSA-2025-03264-TRR

Prof. David A Hume
University of Queensland
Macrophage Biology Research Group
-
Brisbane, Qld 4102
Australia

Dear Dr. Hume,

Thank you for submitting your Research Article entitled "Mutation in the rat interleukin 34 gene impacts macrophage development, homeostasis and inflammation". We appreciate your thorough responses to the remaining points from Reviewer 3. It is a pleasure to let you know that your manuscript is now accepted for publication in Life Science Alliance. Congratulations on this interesting work.

DISTRIBUTION OF MATERIALS:

Again, congratulations on a very nice paper. I hope you found the review process to be constructive and are pleased with how the manuscript was handled editorially. We look forward to future exciting submissions from your lab.

Sincerely,
